# FourierMamba: Fourier Learning Integration with State Space Models for Image Deraining

Dong Li [* 1]  Yidi Liu [* 1]  Xueyang Fu [1]  Jie Huang [1]  Senyan Xu [1]  Qi Zhu [1]  Zheng-Jun Zha [1]

## Abstract

Image deraining aims to remove rain streaks from rainy images and restore clear backgrounds. Currently, some research that employs the Fourier transform has proved to be effective for image deraining, due to it acting as an effective frequency prior for capturing rain streaks. However, despite there exists dependency of low frequency and high frequency in images, these Fourier-based methods rarely exploit the correlation of different frequencies for conjuncting their learning procedures, limiting the full utilization of frequency information for image deraining. Alternatively, the recently emerged Mamba technique depicts its effectiveness and efficiency for modeling correlation in various domains (e.g., spatial, temporal), and we argue that introducing Mamba into its unexplored Fourier spaces to correlate different frequencies would help improve image deraining. This motivates us to propose a new framework termed FourierMamba, which performs image deraining with Mamba in the Fourier space. Owing to the unique arrangement of frequency orders in Fourier space, the core of FourierMamba lies in the scanning encoding of different frequencies, where the low-high frequency order formats exhibit differently in the spatial dimension (unarranged in axis) and channel dimension (arranged in axis). Therefore, we design FourierMamba that correlates Fourier space information in the spatial and channel dimensions with distinct designs. Specifically, in the spatial dimension Fourier space, we introduce the zigzag coding to scan the frequencies to rearrange the orders from low to high frequencies, thereby orderly correlating the connections between frequencies; in the channel dimension Fourier space with arranged orders of frequencies in axis, we can directly use Mamba to perform frequency correlation and improve the channel information representation. Extensive experiments reveal that our method outperforms state-of-the-art methods both qualitatively and quantitatively.

## 1. Introduction

Images taken in rainy conditions exhibit significant degradation in detail and contrast due to rain in the air, leading to unpleasant visual results and the loss of frequency information. This issue can severely impact the performance of outdoor computer vision systems, such as autonomous driving and video surveillance (Wang et al., 2022a). To mitigate the effects of rain, many image deraining methods (Fu et al., 2011; Xiao et al., 2022) have emerged in recent years, aiming to remove rain streaks and restore clear backgrounds in images.

The advent of deep learning has spurred this field forward, with several learning-based deraining methods achieving remarkable success (Fu et al., 2017b; Yang et al., 2017; Zhang & Patel, 2018). Among them, some studies utilize the Fourier transform for deraining in the frequency domain (Zhou et al., 2023; Guo et al., 2022), proving effective. The key insights inspiring the use of the Fourier transform for image deraining are twofold: 1) The Fourier transform can separate image degradation and content components to some extent, serving as a prior for image deraining, as shown in Figure 1; 2) The Fourier domain possesses global properties, where each pixel in Fourier space is involved with all spatial pixels. Thus, it makes sense to explore the task of rain removal using the Fourier transform. However, despite the existence of low frequency and high frequency dependencies in images, previous Fourier-based methods rarely utilize the correlation of different frequencies to combine their learning process. As shown in Figure 1, the commonly used $1 \times 1$ convolutions cannot correlate different frequencies, limiting the full utilization of frequency information in the image. Therefore, we seek to exploit the beneficial properties of the Fourier transform while exploring correlating different frequencies.

*Equal contribution  [1]University of Science and Technology of China, Hefei, China. Correspondence to: Zheng-Jun Zha <zhazj@ustc.edu.cn>.

*Proceedings of the 42$^{nd}$ International Conference on Machine Learning*, Vancouver, Canada. PMLR 267, 2025. Copyright 2025 by the author(s).

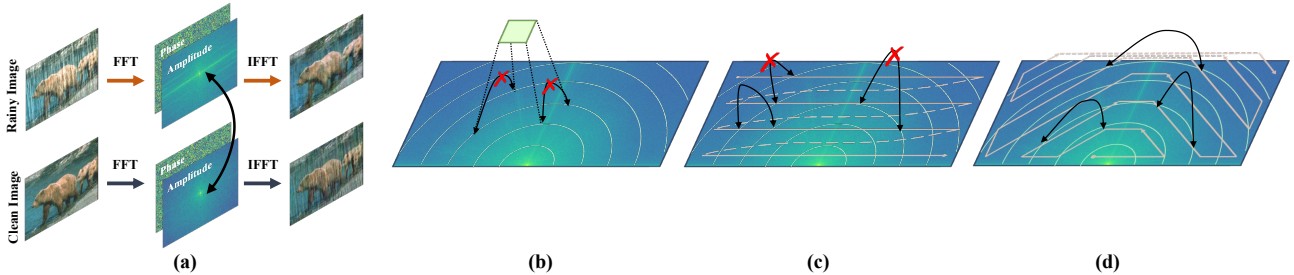

*Figure 1.* Observation and comparison of different frequency modeling methods. (a) Observation of the amplitude spectrum exchange. The degradation is mainly in amplitude components, so the Fourier transform helps to disentangle the image content and rain. (b) The commonly used $1 \times 1$ convolution cannot model the relationship between different frequencies. (c) Previous scanning in Fourier space will fail to establish the ordered dependence between frequencies. (d) Our proposed method achieves ordered frequency dependence from low to high (or vice versa), thus fully utilizing frequency information.

Recently, an improved structured state-space sequence model (S4) with a selective scanning mechanism, Mamba, gives us hope. The selective methodology of Mamba can explicitly build the correlation among image patches or pixels. Recent studies have witnessed the effectiveness and efficiency of Mamba in various domains such as spatial and temporal. Therefore, we believe that introducing Mamba into its unexplored Fourier space to correlate different frequencies will improve image deraining.

In this paper, we propose a novel framework named FourierMamba, which performs image deraining using mamba in the Fourier domain. Following the "spatial interaction + channel evolution" rule that has also been validated on Mamba (Guo et al., 2024; Behrouz et al., 2024), we design the Mamba framework in the Fourier domain from both spatial and channel dimensions. Considering the unique arrangement of frequency orders in the Fourier domain, the core of FourierMamba lies in the scanning encoding of different frequencies, where the low-high frequency order formats unarranged in the spatial axis and arranged in the channel axis. Therefore, our proposed FourierMamba correlates Fourier space information in spatial and channel dimensions with distinct designs.

Specifically, **in the spatial dimension of the Fourier space**, low-high frequencies follow a concentric circular arrangement with lower frequencies near the center and higher frequencies around the periphery. If previous scanning method (Liu et al., 2024) is used directly, the orderliness between frequencies will be destroyed, as shown in Figure 1. We note that the zigzag coding in the JPEG compression field can place lower-frequency coefficients at the forefront of the array, while higher-frequency coefficients are positioned at the end. Hence, we introduce the zigzag coding to scan the frequency in the spatial dimension, rearranging the order from low to high frequency. Due to the symmetry of the frequency orders in the Fourier space, we do not directly

employ the zigzag coding in its originally used space; instead, we implement it in a circling-like manner that matches the symmetric frequency orders in Fourier space. In this way, this method orderly correlates the connections between frequencies, as shown in Figure 1. **In the channel dimension of the Fourier space**, the frequency order is arranged along the axis, following the order of low in the middle to high on both sides. Therefore, we can directly use Mamba for frequency correlation, thus improving channel information representation and enhancing global properties on the channels.

In summary, our contributions are as follows: (1) We propose a novel framework FourierMamba that combines Fourier priors and State Space Model for correlating different frequencies in the Fourier space to enhance image deraining. (2) To rearrange the order from low to high frequency in the spatial dimension Fourier space, we propose a scanning method based on zigzag coding to orderly correlate different frequencies. (3) Based on the channel-dimension Fourier transform, we utilize Mamba to scan on the channels and correlate different frequencies to improve channel information representation. Extensive experiments demonstrate that the proposed FourierMamba surpasses state-of-the-art methods both qualitatively and quantitatively.

## 2. Related Works

**Fourier transform.** Recently, the Fourier Transform has demonstrated its effectiveness in global modeling (Chi et al., 2019; 2020). This transformation converts signals into a domain characterized by global statistical properties, facilitating advancements across various fields (Huang et al., 2022; Lee et al., 2018; Li et al., 2023; Pratt et al., 2017; Xu et al., 2021; Yang & Soatto, 2020). Due to its efficacy in global modeling, the Fourier Transform has been introduced into low-level vision tasks (Fuoli et al., 2021; Mao et al., 2023). As an early attempt, (Fuoli et al., 2021) proposes

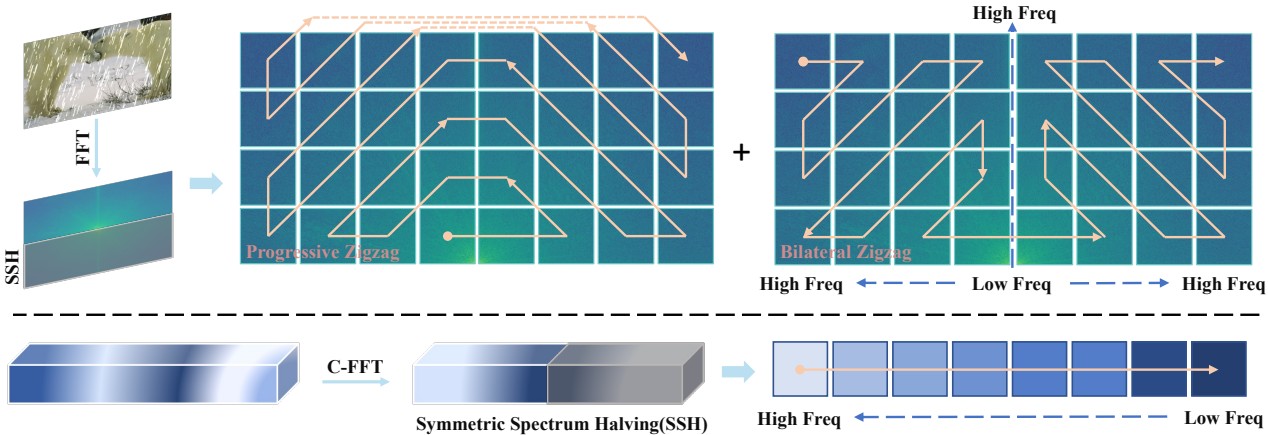

*Figure 2.* Our proposed Fourier space scanning method in the spatial dimension (top) and channel dimension (bottom). For simplicity, only one direction is shown for each scanning method, and in fact each method also performs a scan opposite to that shown.

a Fourier Transform-based loss to optimize global high-frequency information for efficient image super-resolution. DeepRFT (Mao et al., 2023) is proposed for image deblurring, employing a global receptive field to capture both low and high-frequency characteristics of various blurs, a concept similarly applied in image inpainting (Suvorov et al., 2022). FECNet (Huang et al., 2022) demonstrates that the amplitude of Fourier features decouples global luminance components, thereby proving effective for image enhancement. (Yu et al., 2022) observes a similar phenomenon in image dehazing, where the amplitude reflects global haze-related information. In contrast, we introduce a progressive scanning strategy in the Fourier domain, enhancing the global modeling capability while addressing the directional sensitivity issues of visual Mamba.

**State Space Models.** State Space Models (SSMs) have received a lot of attention recently due to their global modeling capabilities as well as linear complexity, with (Gu et al., 2022) initially introducing the base design of SSM models, and (Mehta et al., 2022) further enhancing their performance through gating units.More recently, the performance of Mamba (Gu & Dao, 2023), proposed based on selective scan mechanism and efficient hardware design, has seen significant enhancement. It stands as an efficient alternative to Transformers, finding applications in various domains including image classification (Zhu et al., 2024)(Liu et al., 2024), object detection(Chen et al., 2024), and remote sensing(Zhao et al., 2024).In the field of image restoration, (Guo et al., 2024) (Shi et al., 2024) initially introduced a general restoration framework based on the Mamba module but did not fully exploit the frequency domain information of images. (Sun et al., 2024) introduces a network combining Transformer and Mamba to capture long-range dependencies related to rain. (Yamashita & Ikehara, 2024) achieves effective deraining by parallelizing

frequency-domain processing branches with the Mamba branch. (Zhen et al., 2024) introduced a wavelet transform branch, yet the scanning in the wavelet domain fails to fully extract global frequency domain information. This paper proposes a novel Mamba restoration network based on Fourier transform, aiming to comprehensively exploit the frequency domain information of images. Please see the Appendix for more related works.

## 3. Methodology

### 3.1. Preliminary

**Fourier transform.** Fourier transform is a widely used technique for analyzing the frequency content of an image. For images with multiple color channels, the Fourier transform is applied to each channel separately. Given an image $X \in \mathbb{R}^{H \times W \times C}$, the Fourier transform $\mathcal{F}$ converts it to Fourier space as the complex component F(x), which is expressed as:

$$\mathcal{F}(x)(u,v) = \frac{1}{\sqrt{HW}} \sum_{h=0}^{H-1} \sum_{w=0}^{W-1} x(h,w) e^{-j2\pi\left(\frac{h}{H}u + \frac{w}{W}v\right)}, \tag{1}$$

where $u$ and $v$ indicate the coordinates of the Fourier space. $\mathcal{F}^{-1}(x)$ defines the inverse Fourier transform accordingly. Both the Fourier transform and its inverse procedure can be efficiently implemented using FFT/IFFT algorithms (Frigo & Johnson, 1998). The amplitude component $\mathcal{A}(x)(u,v)$ and phase component $\mathcal{P}(x)(u,v)$ are expressed as:

$$\mathcal{A}(x)(u,v) = \sqrt{R^2(x)(u,v) + I^2(x)(u,v)},$$
$$\mathcal{P}(x)(u,v) = \arctan\left[\frac{I(x)(u,v)}{R(x)(u,v)}\right], \tag{2}$$

where $R(x)(u,v)$ and $I(x)(u,v)$ represent the real and

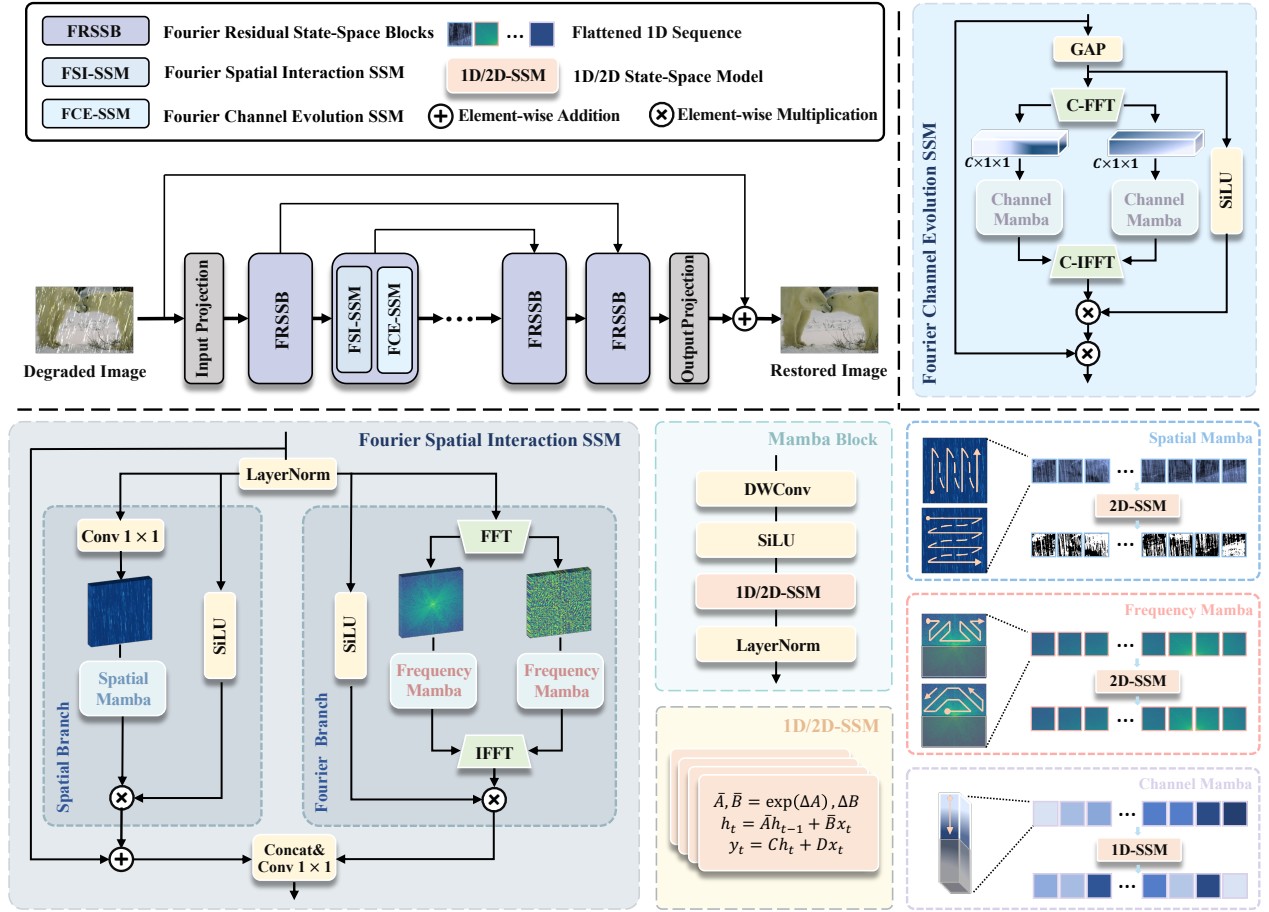

*Figure 3.* The overall architecture of the FourierMamba. Our FourierMamba consists of multiscale hierarchical design Fourier Residual State-Space Blocks(FRSSB). The core modules of FRSSB are Fourier Spatial Interaction SSM(FSI-SSM) and Fourier Channel Evolution SSM(FCE-SSM).

imaginary parts respectively. The Fourier transform and its inverse procedure are applied independently to each channel of the feature maps.

**Channel-dimension Fourier transform.** We introduce the channel-dimension Fourier transform (C-FFT) by individually applying the Fourier transform along the channel dimension for each spatial position. For each position $(h \in \mathbb{R}^{H-1}, w \in \mathbb{R}^{W-1})$ within $X \in \mathbb{R}^{H \times W \times C}$, denoted as $x(h, w, 0 : C - 1)$ and abbreviated as $y(0 : C - 1)$, Fourier transform $\mathcal{F}(\cdot)$ converts it to Fourier space as the complex component $\mathcal{F}(y)$, which is expressed as:

$$\mathcal{F}(y(0 : C-1))(z) = \frac{1}{C} \sum_{c=0}^{C-1} y(c) e^{-j2\pi \frac{c}{C} z}, \quad (3)$$

.

Similarly, the amplitude component $\mathcal{A}(y(0 : C-1))(z)$ and phase component $\mathcal{P}(y(0 : C-1))(z)$ of $\mathcal{F}(y(0 : C-1))(z)$

are expressed as:

$$
\begin{aligned}
&\mathcal{A}(y(0 : C-1))(z) \\
&= \sqrt{R^2\left(y(0 : C-1)\right)(z) + I^2\left(y(0 : C-1)\right)(z)}, \\
&\mathcal{P}(y(0 : C-1))(z) \\
&= \arctan\left[\frac{I\left(y(0 : C-1)\right)(z)}{R\left(y(0 : C-1)\right)(z)}\right].
\end{aligned}
$$

These operations can also be applied for the global vector derived by the pooling operation. In this way, $\mathcal{A}(z)$ and $\mathcal{P}(z)$ signify the magnitude and directional changes in the magnitude of various channel frequencies, respectively. Both of these metrics encapsulate global statistics related to channel information.

**State Space Models.** State Space Models (SSMs) serve as the cornerstone for transforming one-dimensional inputs into outputs through latent states, utilizing a framework of linear ordinary differential equations. Mathematically, SSMs can be formulated as follows, representing linear

ordinary differential equations (ODEs):

$$h'(t) = \boldsymbol{A}h(t) + \boldsymbol{B}x(t),$$
$$y(t) = \boldsymbol{C}h(t) + \boldsymbol{D}x(t),$$
(4)

where, $h(t) \in \mathbb{R}^N$ denotes the hidden state vector, where N represents the size of the state. The parameters $\boldsymbol{A} \in \mathbb{R}^{N \times N}$, $\boldsymbol{B} \in \mathbb{R}^N$, and $\boldsymbol{C} \in \mathbb{R}^N$ are associated with the state size N, while $\boldsymbol{D} \in \mathbb{R}^1$ represents the skip connection.

Discrete versions of these models, such as Mamba(Gu & Dao, 2023), include a discretization step via the zero-order hold (ZOH) method. This enables the models to adaptively scan and adjust to the input data using a selective scanning mechanism. This mechanism provides a global receptive field with linear complexity, which is advantageous for image restoration tasks.

### 3.2. Scanning in Fourier Space

Despite the unique characteristics of the selective scan mechanism (S6), it processes input data causally. Given the non-causal nature of visual data, directly applying this strategy to patches and flat images fails to estimate relations with unscanned patches, leading to a "directional sensitivity" issue constrained by the acceptance domain. Numerous methods have attempted to tackle this problem in the spatial domain (Liu et al., 2024; Guo et al., 2024). However, for image restoration, the Fourier space and its associated priors are crucial. Hence, we explore addressing the "directional sensitivity" issue within this domain. Specifically, we customize Fourier scanning strategies from both spatial and channel dimensions.

For the **spatial dimension**, each pixel point in the Fourier space contains global information, with its frequencies distributed in concentric circles. Scanning methods based on spatial arrangements (Liu et al., 2024) disrupt the high-low frequency relationships in the frequency domain, thus hindering the modeling of image degradation information.

Therefore, we aim to devise a scanning method in the Fourier space to progressively model the frequency characteristics of images. An intuitive approach is to calculate the Euclidean distance from each point in the spectrum to the center point. On the shifted Fourier spectrum, the smaller the distance to the center point, the lower the frequency. The flaw of this intuitive approach is that for images of different sizes, it requires recalculating the Euclidean distance from each point to the center point. The additional computational overhead introduced by this flaw makes this approach impractical in the field of image restoration.

In JPEG compression, zigzag coding is commonly used among the Discrete Cosine Transform (DCT) coefficients of JPEG, where it prioritizes the energy-concentrated low-frequency coefficients at the beginning of the array, and

places the less significant high-frequency coefficients towards the end, thereby facilitating more effective compression. Inspired by compression algorithms, we introduce a method that adopts the zigzag coding approach to scan the magnitude and phase spectra.

Additionally, due to the symmetry of the two-dimensional Fourier transform, scanning the entire spectrum would disrupt the symmetry in the Fourier space, potentially leading to the collapse of network optimization. Therefore, we scan half of the spectrum and then deduce the other half based on the central symmetry of the amplitude and the anti-central symmetry of the phase.

Specifically, we design two scanning strategies, as illustrated in the Figure 2. The first scanning method employs a dual zigzag pattern named **bilateral zigzag**, starting from the vertex of the highest frequency on one side of the spectrum, progressing in a zigzag pattern toward the center's low frequencies; similarly, it then zigzags to the opposite side's highest frequency. This scanning approach not only models the association between high and low frequencies but also takes into account the periodicity of the Fourier spectrum. Due to the periodic nature of the Fourier transform, the high-frequency ends on either side should, in fact, be contiguous. The second method builds upon the low-to-high frequency sequence established by zigzag scanning and conducts a scan from low to high frequencies, which is named **progressive zigzag**. This method is motivated by the tendency of neural networks to initially learn low-frequency information when extracting image characteristics. Following the previous method (Liu et al., 2024; Guo et al., 2024), we reverse the above two scanning methods as additional scanning directions.

For the **channel dimension** Fourier space, since it is a one-dimensional sequence arranged in order of low to high frequencies, we directly scan it one-dimensionally. Similarly, due to the symmetry of the Fourier transform, we scan only half and derive the other half. Through Fourier space scanning in both spatial and channel dimensions, we can correlate the connections between frequencies in an orderly manner, thereby making full use of frequency information to improve rain removal.

### 3.3. FourierMamba

#### 3.3.1. OVERALL FRAMEWORK

In Figure 3, we illustrate our proposed FourierMamba. Given a rainy image $I \in \mathbb{R}^{H \times W \times 3}$, FourierMamba first uses $3 \times 3$ convolution layers to generate shallow features with dimensions of $H \times W \times C$, where $H$ and $W$ represent height and width, and $C$ denotes the number of channels. Subsequently, we employ a multi-scale U-Net architecture to obtain deep features. This stage consists of a stack of

| Input | GT | Classical | Bilateral | Progressive | Ours |
|-------|-----|-----------|-----------|-------------|------|

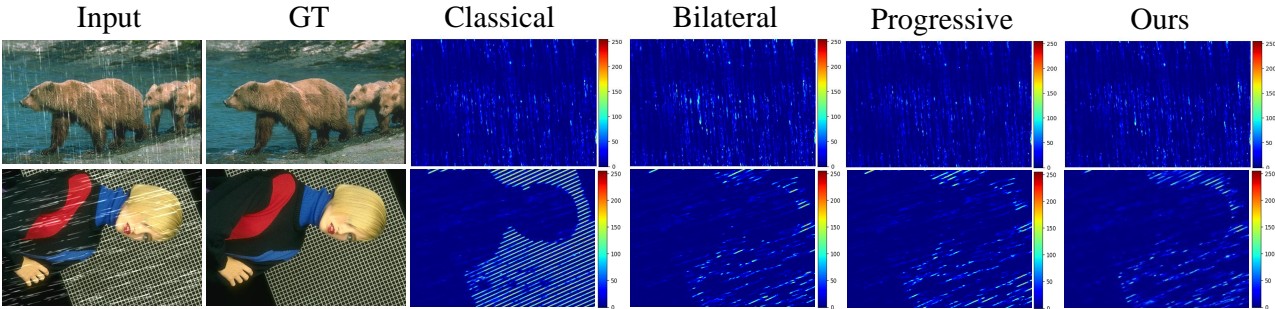

*Figure 4.* The error map between the GT and the restored images using various scanning methods in Fourier space. The two scanning methods we propose can achieve smaller errors than using classical scanning method (Liu et al., 2024). And the combination of the two scanning methods is better than either one.

Fourier Residual State-Space Groups, each containing several Fourier Residual State-Space Blocks (FRSSB). The FRSSB incorporates our two core designs: the Fourier Spatial Interaction SSM block and the Fourier Channel Evolution SSM block.

### 3.3.2. FOURIER SPATIAL INTERACTION SSM

The structure of the Fourier Spatial Interaction State Space Model (FSI-SSM) is shown in Figure 3. We first apply LayerNorm to transform the input features $F_{in}$ into $F_l$. To facilitate the interaction between spatial and frequency information, FSI-SSM employs both a Fourier branch and a spatial branch to collaboratively process $F_{in}$.

**Fourier Branch:** $F_l$ is transformed into the Fourier spectrum through the Fast Fourier Transform, subsequently decomposed into the amplitude spectrum $\mathcal{A}(F_l)$ and phase spectrum $\mathcal{P}(F_l)$. The amplitude spectrum and phase spectrum are then processed separately using the progressive frequency scanning method illustrated in Figure 2 to obtain $\mathcal{A}'(F_l)$ and $\mathcal{P}'(F_l)$.

$$\begin{aligned} \mathcal{A}'(F_l) = \mathrm{FourScan}(\mathcal{A}(F_l)), \\ \mathcal{P}'(F_l) = \mathrm{FourScan}(\mathcal{P}(F_l)), \end{aligned} \quad (5)$$

where $\mathrm{FourScan}$ is the sequence transformation using the Fourier space scan described in Sec. 3.2. Following a series of works (Liu et al., 2024; Guo et al., 2024; Zhen et al., 2024), the sequence transformation employs the following operation sequence: $DWConv \rightarrow SiLU \rightarrow SSM \rightarrow LayerNorm$. We then perform an inverse Fourier transform on the processed spectrum and multiply it with the output of SiLU.

$$F_f = (\mathcal{F}^{-1}(\mathcal{A}'(F_l), \mathcal{P}'(F_l))) \odot \mathrm{SiLU}(F_l), \quad (6)$$

where $F_f$ is the output of the fourier branch, and $\odot$ is the Hadamard product.

**Spatial Branch** In the spatial domain, we feed the input features $F_l$ into two parallel sub-branches. One sub-branch activates the features using the SiLU function. The other sub-branch performs spatial Mamba on features after $1 \times 1$ convolution. Specifically, spatial Mamba adopts the same operation sequence as the above frequency branch but the scanning in SSM uses the two-dimensional selective scanning module shown in Figure 3, which follows previous work (Liu et al., 2024; Guo et al., 2024). Finally, the outputs of the two sub-branches are multiplied element-wise to obtain the output $F_s$.

$$F_s = \mathrm{SpaScan}(\mathrm{Conv}(F_l)) \odot \mathrm{SiLU}(F_l), \quad (7)$$

where $\mathrm{Conv}$ is $1 \times 1$ convolution and $\mathrm{SpaScan}$ is the spatial Mamba mentioned above. Subsequently, we employ a residual connection to add the spatial output to $F_{in}$. The spatial branch captures global features in the spatial domain which complement the frequency correlations captured by the Fourier branch in the frequency domain, thereby benefiting the performance of image deraining. Hence, we concatenate the outputs of the spatial and frequency branches and use a $1 \times 1$ convolution for the fusion of spatial and frequency information.

### 3.3.3. FOURIER CHANNEL EVOLUTION SSM

Previous work (Guo et al., 2024) claims that selecting key channels can avoid channel redundancy in SSM. Since each channel contains the information of all channels after the channel-dimension Fourier transform (C-FFT), we perform channel interaction in the Fourier domain to efficiently correlate different frequencies of channels. As depicted in Figure 3, our proposed Fourier Channel Evolution SSM (FCE-SSM) consists of three sequential parts: applying the Fourier transform along the channel dimension to obtain channel-wise Fourier domain features, scanning its amplitude and phase, then restoring to the spatial domain. Specifically, assuming the input features are denoted as $F_r \in \mathbb{R}^{H_r \times W_r \times C_r}$, we first perform global average pooling on it.

$$F_g = \frac{1}{H_r W_r} \sum_{h=0}^{H_r-1} \sum_{w=0}^{W_r-1} F_g(h, w), \quad (8)$$

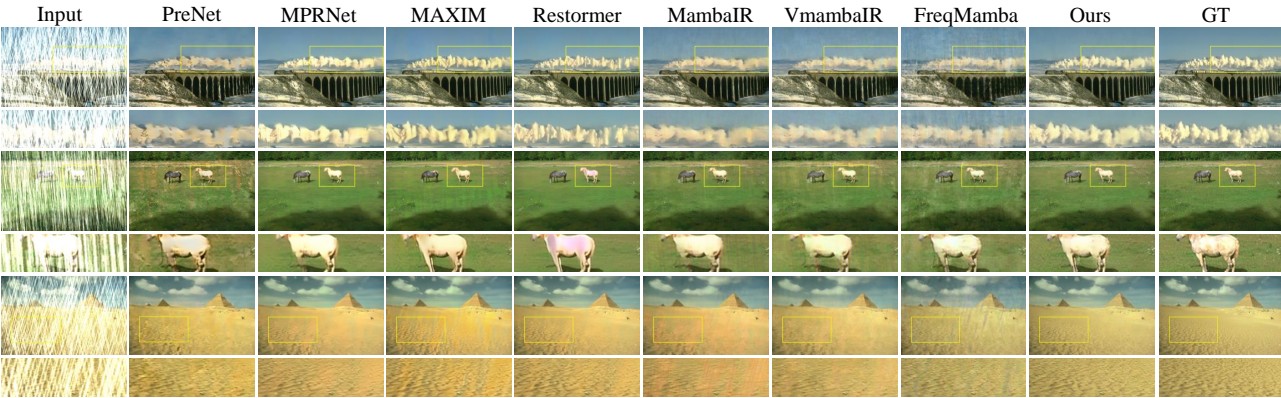

*Figure 5.* Qualitative comparison on Rain100H (Yang et al., 2017). Zoom in for better visualization.

where $F_g \in \mathbb{R}^{1 \times 1 \times C_r}$ corresponds to the center point of the amplitude spectrum of $F_r$ (see supplementary material), which effectively encapsulates the global information of the feature. Then, we use the channel-dimensional Fourier transform shown in Equ. 3 on $F_g$ to obtain $\mathcal{F}(F_g)(z)$. Based on this, we use Equ. 4 for $\mathcal{F}(F_g)(z)$ to obtain its amplitude component $\mathcal{A}(F_g)(z)$ and phase component $\mathcal{P}(F_g)(z)$. Since the amplitude spectrum and phase spectrum have obvious information meaning, we choose to perform Mamba scanning on these two components.

$$\begin{aligned} \mathcal{A}(F_g)(z)' &= \text{ChaScan}(\mathcal{A}(F_g)(z)), \\ \mathcal{P}(F_g)(z)' &= \text{ChaScan}(\mathcal{P}(F_g)(z)), \end{aligned} \quad (9)$$

where ChaScan is a one-dimensional sequence transformation that uses the following sequence of operations: $DWConv \rightarrow SiLU \rightarrow SSM \rightarrow LayerNorm$. Its scanning method is shown in Figure 2. After the Mamba correlates different frequencies in the channel dimension, we perform an inverse Fourier transform on it and multiply the result with the channel features after SiLU activation.

$$F_a = (\mathcal{F}^{-1}(\mathcal{A}(F_g)(z)', \mathcal{P}(F_g)(z)')) \odot \text{SiLU}(F_g), \quad (10)$$

where $F_a \in \mathbb{R}^{1 \times 1 \times C_r}$ is the channel feature after correlating different frequencies. Finally, we multiply it with the spatial feature in a form of attention to get the output $F_c \in \mathbb{R}^{H_r \times W_r \times C_r}$.

$$F_c = F_a \odot F_r. \quad (11)$$

### 3.3.4. OPTIMIZATION

We impose constraints in both the spatial and frequency domains. In the spatial domain, we utilize the L1 loss between the final output $Y_{out}$ and the ground truth $Y_{gt}$. In the frequency domain, we apply the L1 loss based on the Fourier transform. The overall loss function is formulated as follows:

$$\mathcal{L}_{total} = \|Y_{out} - Y_{gt}\|_1 + \lambda \|\mathcal{F}(Y_{out}) - \mathcal{F}(Y_{gt})\|_1, \quad (12)$$

where $\lambda$ is the balancing weight. In particular, $\lambda$ is set to 0.02 empirically.

## 4. Experiment

### 4.1. Experimental Settings

**Datasets.** For training, we employ the widely used Rain13k dataset (Chen et al., 2021). It contains 13,712 image pairs in the training set, and we evaluate the results on Rain100H (Yang et al., 2017), Rain100L (Yang et al., 2017), Test2800 (Fu et al., 2017b), and Test1200 (Zhang & Patel, 2018).

**Evaluation Metrics.** Following previous work (Zamir et al., 2021; 2022), we adopt two commonly used quantitative metrics for evaluations: Peak Signal-to-Noise Ratio (PSNR) (Huynh-Thu & Ghanbari, 2008) and Structural Similarity Index (SSIM) (Wang et al., 2004).

**Implementation Details.** Our model is implemented within the PyTorch framework and executed on an NVIDIA A100 GPU. The number of blocks per layer has an impact on both the model's parameter count and its deraining performance. After balancing the weights, we configure the blocks per layer as [2, 3, 3, 4, 3, 3, 2], which allows us to achieve commendable performance with a reasonable number of parameters. We adopt the progressive training strategy. Specifically, we set the total number of iterations to 80,000 and image sizes to [160, 256, 320, 384], with the corresponding batch sizes of [8, 4, 2, 1]. We utilize the Adam optimizer with default parameters. The initial learning rate is established at $3 \times e^{-4}$, followed by a gradual decay to $1 \times e^{-6}$ using a cosine annealing schedule.

### 4.2. Comparison with State-of-the-art Methods

**Comparison on Benchmark Datasets.** We first verify the effectiveness of FourierMamba through training models on a mixture of synthetic datasets. We compare our method

*Table 1.* Quantitative comparison (PSNR/SSIM) for Image Deraining on five benchmark datasets. The highest and second-highest performances are marked in bold and underlined. '-' indicates the result is not available.

| Method | Venue | Rain100H (Yang et al., 2017) | | Rain100L (Yang et al., 2017) | | Test2800 (Fu et al., 2017b) | | Test1200 (Zhang & Patel, 2018) | | Param(M) | GFlops |
|---|---|---|---|---|---|---|---|---|---|---|---|
| | | PNSR ↑ | SSIM ↑ | PNSR ↑ | SSIM ↑ | PNSR ↑ | SSIM ↑ | PNSR ↑ | SSIM ↑ | | |
| DerainNet (Fu et al., 2017b) | TIP'17 | 14.92 | 0.592 | 27.03 | 0.884 | 24.31 | 0.861 | 23.38 | 0.835 | 0.058 | 1.453 |
| UMRL (Yasarla & Patel, 2019) | CVPR'19 | 26.01 | 0.832 | 29.18 | 0.923 | 29.97 | 0.905 | 30.55 | 0.910 | 0.98 | - |
| RESCAN (Li et al., 2018) | ECCV'18 | 26.36 | 0.786 | 29.80 | 0.881 | 31.29 | 0.904 | 30.51 | 0.882 | 1.04 | 20.361 |
| PreNet (Ren et al., 2019) | CVPR'19 | 26.77 | 0.858 | 32.44 | 0.950 | 31.75 | 0.916 | 31.36 | 0.911 | 0.17 | 73.021 |
| MSPFN (Jiang et al., 2020) | CVPR'20 | 28.66 | 0.860 | 32.40 | 0.933 | 32.82 | 0.930 | 32.39 | 0.916 | 13.22 | 604.70 |
| SPAIR (Purohit et al., 2021) | ICCV'21 | 30.95 | 0.892 | 36.93 | 0.969 | 33.34 | 0.936 | 33.04 | 0.922 | - | - |
| MPRNet (Zamir et al., 2021) | CVPR'21 | 30.41 | 0.890 | 36.40 | 0.965 | 33.64 | 0.938 | 32.91 | 0.916 | 3.64 | 141.28 |
| Restormer (Zamir et al., 2022) | CVPR'22 | 31.46 | 0.904 | 38.99 | 0.978 | 34.18 | 0.944 | 33.19 | 0.926 | 24.53 | 174.7 |
| Fourmer (Zhou et al., 2023) | ICML'23 | 30.76 | 0.896 | 37.47 | 0.970 | - | - | 33.05 | 0.921 | 0.4 | 16.753 |
| IR-SDE (Luo et al., 2023a) | ICML'23 | 31.65 | 0.904 | 38.30 | 0.980 | 30.42 | 0.891 | - | - | 135.3 | 119.1 |
| MambaIR (Guo et al., 2024) | arxiv'24 | 30.62 | 0.893 | 38.78 | 0.977 | 33.58 | 0.927 | 32.56 | 0.923 | 31.51 | 80.64 |
| VMambaIR (Shi et al., 2024) | arxiv'24 | 31.66 | 0.909 | 39.09 | 0.979 | 34.01 | 0.944 | 33.33 | 0.926 | - | - |
| FreqMamba (Zhen et al., 2024) | arxiv'24 | 31.74 | 0.912 | 39.18 | 0.981 | **34.25** | **0.951** | 33.36 | 0.931 | 14.52 | 36.49 |
| FourierMamba(Ours) | - | **31.79** | **0.913** | **39.73** | **0.986** | 34.23 | 0.949 | **34.76** | **0.938** | 17.62 | 22.56 |

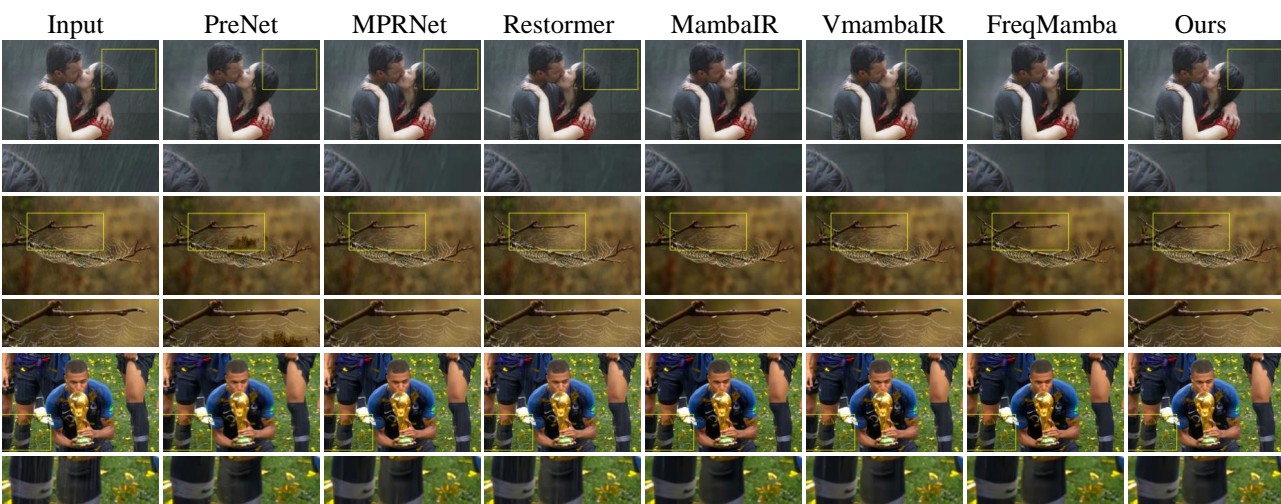

*Figure 6.* Qualitative comparison of real-world rainy images from Internet-Data (Wang et al., 2019).

with these deraining methods: DerainNet (Fu et al., 2017a), UMRL (Yasarla & Patel, 2019), RESCAN (Li et al., 2018), PreNet (Ren et al., 2019), MSPFN (Jiang et al., 2020) , SPAIR (Purohit et al., 2021) , MPRNet (Zamir et al., 2021) , Restormer(Zamir et al., 2022), Fourmer (Zhou et al., 2023), IR-SDE (Luo et al., 2023b), MambaIR (Guo et al., 2024) VMambaIR(Shi et al., 2024) and FreqMamba(Zhen et al., 2024). Table 1 reports the performance evaluation on four datasets. It can be seen that our method achieves the best performance on most datasets, which emphasizes the effectiveness of FourierMamba in improving deraining performance.

To demonstrate the enhanced fidelity and detail levels exhibited by the images generated by our proposed FourierMamba, we compare the visual quality of challenging degraded images from the Rain100H dataset in Figure 5. Our method achieves excellent results when faced with complex or extremely severe rain streaks. Compared to previous methods, our FourierMamba achieves impeccable performance in both global and local restoration. For instance, by

zooming into the red boxed area in Figure 5, our method removes more rain streak residues while better restoring texture details.

**Real-world Deraining Transferred from Synthetic Datasets.**

To verify the generalization of the proposed method in real-world scenarios, we use the model trained on Rain13k to examine the real-world deraining capabilities. We evaluate the model trained on the synthetic dataset on the real-world dataset Inrernet-Data (Wang et al., 2019) without ground truth. As shown in Figure 6, FourierMamba is able to remove these complex rains and restore the clean background. In contrast, other deraining methods do not handle the effect of rain cleanly. More generalization results in real-world scenarios can be found in the Appendix.

**Training On Real-world Rainy Datasets.**

To further explore the potential of the proposed method, we use the real-world dataset SPAData (Wang et al., 2019) to train FourierMamba. In Table 2, our method is compared

*Table 2.* Quantitative comparison of training and testing on the real-world dataset SPA-Data.

| Method | RESCAN | PReNet | SPDNet | DualGCN | Restormer | DRSformer | Ours |
|---|---|---|---|---|---|---|---|
| PSNR | 38.11 | 40.16 | 43.20 | 44.18 | 47.98 | 48.54 | **49.18** |
| SSIM | 0.9707 | 0.9816 | 0.9871 | 0.9902 | 0.9921 | 0.9924 | **0.9931** |

*Table 3.* Ablation studies of key designs in the proposed method.

| | w/o FSI-SSM | w/o FCE-SSM | w/o SDF | w/o CDF | Ours |
|---|---|---|---|---|---|
| PSNR | 39.05 | 39.08 | 38.25 | 38.72 | **39.73** |
| SSIM | 0.9835 | 0.9836 | 0.9810 | 0.9827 | **0.9856** |

*Table 4.* Ablation of different scanning methods in Fourier space.

| | Classic(Liu et al., 2024) | Bilateral | Progressive | Ours |
|---|---|---|---|---|
| PSNR | 38.82 | 39.31 | 39.28 | **39.73** |
| SSIM | 0.9817 | 0.9844 | 0.9843 | **0.9856** |

with these methods RESCAN (Li et al., 2018), PReNet (Ren et al., 2019), SPDNet (Yi et al., 2021), DualGCN (Fu et al., 2021), Restormer (Zamir et al., 2022), and DRSformer (Chen et al., 2023a) with the same experimental settings. Surprisingly, we observe that FourierMamba acquires significant real-world rain removal capabilities. This shows that our method can effectively learn the precipitation model of real rain.

### 4.3. Ablation Studies

We perform ablations on the key designs and scanning methods of the framework on the Rain100L.

**Fourier Spatial Interaction SSM (FSI-SSM) and Fourier Channel Evolution SSM (FCE-SSM).** We replace the mamba scan in FSI-SSM and FCE-SSM with $1 \times 1$ convolution, called w/o FSI-SSM and w/o FCE-SSM respectively. It can be seen from Table 3 that since $1 \times 1$ convolution cannot model the dependence of different frequencies, its performance is worse than the mamba scan in the Fourier domain in both the spatial dimension and the channel dimension.

**Fourier prior.** We do not use Fourier transform in the spatial dimension and channel dimension respectively, but directly perform mamba scanning, which are called without spatial dimension Fourier (w/o SDF) and without channel dimension Fourier (w/o CDF) respectively. It can be seen from Table 3 that after losing the Fourier prior in the spatial dimension and channel dimension, the performance drops significantly. This proves the effectiveness of Fourier prior for removing rain from images. The Fourier prior also improves the visual effect; please refer to the Appendix.

**Scanning method in Fourier space.** We compare several scanning methods of the spatial dimension Fourier space, with the same amount of calculation. Table 4 illustrates that the performance of the two scanning methods we proposed is better than the classic two-dimensional scanning method (Liu et al., 2024). And thanks to complementarity, the combination of the two methods can also further improve performance. The visual comparison in Figure 4 supports this.

## 5. Conclusion

In this paper, we propose a novel image deraining framework, FourierMamba, which utilizes mamba to correlate frequencies in the Fourier space, thus fully exploiting frequency information. Specifically, we design the mamba framework by integrating the unique arrangement of frequency orderings within the Fourier domain across spatial and channel dimensions. In the spatial dimension, we devise two zigzag-based methods to scan frequencies, systematically correlating them. In the channel dimension, due to the ordered arrangement of frequencies along the axis, we directly apply mamba for frequency correlation. This work introduces a novel strategy to address the underutilization of frequency information in image deraining that affects performance. Extensive experiments on multiple benchmarks validate the effectiveness of the proposed method.

## Impact Statement

Due to uncontrollable weather conditions, image acquisition systems inevitably suffer interference from rain. Images captured during rainy conditions experience a significant decline in the quality of object details and contrast due to rain present in the air. Images tainted by rain can also severely impact the performance of outdoor computer vision systems, including autonomous driving and video surveillance. Therefore, image deraining itself holds significant research and application value. Our proposed FourierMamba combines the priors of Fourier space and the correlation modeling capability of Mamba, enabling the network to tackle more complex image deraining tasks. However, from a societal perspective, negative consequences might also follow. For instance, over-reliance on image deraining technology could introduce deviations from actual image textures, affecting effective judgment in autonomous driving and video surveillance. In these cases, it is necessary to combine expert knowledge to make rational decisions.

## Acknowledgement

This work is supported by the National Natural Science Foundation of China (NSFC) under Grants 62225207, 62436008, 62422609 and 62276243.

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

# A. Appendix

## A.1. Limitation

In this work, we introduce FourierMamba and extensively validate its efficacy for image deraining through experiments. Our experiments primarily leverage the widely used U-shaped architecture. We plan to further validate the effectiveness of combining Fourier priors with Mamba on more architectures, such as isotropic and multi-stage architecture.

Furthermore, given the proven priors of Fourier transform for capturing rain streaks, we choose to first validate FourierMamba on image deraining. Our work could also offer novel insights for other low-level vision fields, though it may necessitate integrating priors tailored to the distinct differences between various low-level tasks. Given the universal need across various low-level tasks for Fourier priors and the importance of correlating frequencies, the performing improvements can be positively anticipated. We will explore applications in other low-level tasks in our future work.

## A.2. More related works

**Image deraining.** Traditional image deraining methods focus on separating rain components by utilizing meticulously designed priors, such as Gaussian Mixture Models (Li et al., 2016), Sparse Representation Learning (Gu et al., 2017; Fu et al., 2011), and Directional Gradient Priors (Ran et al., 2020). Although these methods are insightful, they often struggle to cope with complex precipitation patterns and the diverse real-world scenarios. The advent of deep learning has heralded a new era for image deraining. (Fu et al., 2017b) introduces pioneering deep residual networks for image deraining. The initiation of CNNs marked a significant advancement, facilitating more nuanced and adaptive processing of rain streaks across a vast array of images (Yang et al., 2017; Zhang & Patel, 2018). With the evolution of transformers, the development of architectures that incorporate attention mechanisms (Valanarasu et al., 2022; Wang et al., 2022b) has further refined the capacity to recognize and eliminate rain components, addressing previous shortcomings in model generalization and detail preservation. COIC (Ran et al., 2024) presents a Context-based Instance-level Modulation mechanism integrated with rain-/detail-aware contrastive learning to enhance CNN and Transformer models for improved image deraining on mixed datasets. (Hsu & Chang, 2023) proposes a wavelet approximation-aware residual network, which efficiently removes rain from low-frequency structures and high-frequency details at each level separately. In this work, we propose a novel baseline with a block based on Fourier and Mamba to enhance deraining performance.

## A.3. Inference time of the model

In this section, we compare the inference time of the proposed method with several state-of-the-art methods. The comparison results of the model inference time using $512 \times 512$ images on NVIDIA RTX 4090 GPU are shown in Table 5. It can be seen that the inference time of our model is comparable to that of other methods.

*Table 5.* Runtime comparison between our method and other approaches.

| Method | MambaIR | VmambaIR | FreqMamba | Restormer | Ours |
|--------|---------|----------|-----------|-----------|------|
| Runtime (s) | 0.534s | 0.423s | 1.837s | 0.253s | 0.523s |

## A.4. Results on test100

In this section, we add some performance comparisons with other methods on Test100 as shown in Table 6. All methods are trained on rain13k and then tested on Test100. It can be seen that our method still achieves excellent deraining performance.

*Table 6.* Performance comparison on Test100. PSNR ($\uparrow$) and SSIM ($\uparrow$) are reported.

| Metric | PReNet | MPRNet | Restormer | MambaIR | VmambaIR | FreqMamba | Ours |
|--------|--------|--------|-----------|---------|----------|-----------|------|
| PSNR | 24.81 | 30.27 | 32.00 | 31.82 | 31.84 | 31.89 | 32.07 |
| SSIM | 0.851 | 0.897 | 0.923 | 0.922 | 0.918 | 0.921 | 0.925 |

## A.5. Ablation studies and computational overhead

To further demonstrate the effectiveness of Mamba, we present the impact of computational overhead in the first ablation study. For the ablation of FSI-SSM, we compress our model by reducing the number of channels and blocks, achieving a

computational cost similar to that of the "w/o FSI-SSM" variant. The comparison is shown in Table 7. As observed, the model with FSI-SSM still achieves better performance. For the ablation of FCE-SSM, the computational overhead of the variant without FCE-SSM (w/o FCE-SSM) in Table 3 is similar to that of the model with FCE-SSM. The "w/o FCE-SSM" variant stacks several $1 \times 1$ convolutions with residual connections to match the parameter count of Mamba. The specific computational overhead and performance are shown in Table 8. It is evident that, with a similar parameter count, our method outperforms the "w/o FCE-SSM" variant.

*Table 7.* The computational overhead of the ablation study on FSI-SSM.

| Method | PSNR | SSIM | Flops(G) | Params(M) |
|---|---|---|---|---|
| w/o FSI-SSM | 39.05 | 0.9835 | 14.42 | 10.82 |
| **Ours** | 39.37 | 0.9845 | 14.64 | 10.12 |

*Table 8.* The computational overhead of the ablation study on FCE-SSM.

| Method | PSNR | SSIM | Flops(G) | Params(M) |
|---|---|---|---|---|
| w/o FCE-SSM | 39.08 | 0.9836 | 21.08 | 17.81 |
| **Ours** | 39.73 | 0.9856 | 22.56 | 17.62 |

### A.6. Reasons for using channel-dimensional Fourier

To address the limitation of Fourier transform not accounting for channel evolution, we introduce channel-dimension Fourier transform. A pivotal motivation is due to different channels often displaying varying properties of degradation information, which also determine the global information of the image when conjunct different channels. A comparable deduction can be drawn from style transfer research, where the Gram matrix signifies global style information (Li et al., 2017). This inspires us to employ Fourier transform on the channel dimension to enrich the representation of global information.

### A.7. The Relationship between Global Average Pooling and Fourier Transform

We believe that the global average pooling equals $\mathcal{A}(0, 0)$ in the amplitude. In the Appendix, we further verify this. Typically, the Spatail Fourier transform is expressed as:

$$\mathcal{F}(x)(u, v) = \frac{1}{\sqrt{HW}} \sum_{h=0}^{H-1} \sum_{w=0}^{W-1} x(h, w) e^{-j2\pi\left(\frac{h}{H}u + \frac{w}{W}v\right)}. \tag{13}$$

The center point of the amplitude spectrum means that $u$ and $v$ are 0. The formula is as follows:

$$\mathcal{F}(x)(0, 0) = \frac{1}{\sqrt{HW}} \sum_{h=0}^{H-1} \sum_{w=0}^{W-1} x(h, w). \tag{14}$$

It can be seen that the above formula is essentially to find the average value of the entire feature map. Therefore, global average pooling (GAP) is equivalent to taking the center point of the amplitude spectrum.

### A.8. Performance on other low-level vision tasks

To further demonstrate the effectiveness of our approach, we investigate the performance of our model on other low-level vision tasks. Following FreqMamba (Zhen et al., 2024), we evaluate our method on low-light enhancement and image dehazing. We use the LOL-V1 (Wei et al., 2018) and LOL-V2-synthetic (Wei et al., 2018) datasets to evaluate the performance of our method on low-light enhancement, and the Dense-Haze (Ancuti et al., 2019) and NH-HAZE (Ancuti et al., 2020) datasets are used to evaluate the performance of our method on real-world image dehazing. The results for low-light enhancement are shown in the Table 9. The comparison results for image dehazing are presented in the Table 10. It can be seen that our method also demonstrates significant potential for other image restoration tasks.

### A.9. Differences between the proposed method and FreqMamba

Our method focuses on customized design based on the characteristics of Fourier space, combining Fourier priors with state space models and exploring the potential of introducing Mamba directly in the Fourier domain. In contrast, FreqMamba

*Table 9.* Comparison of methods on LOL-V1 and LOL-V2-Syn datasets.

| Method | LOL-V1 | | LOL-V2-Syn | |
|---|---|---|---|---|
| | PSNR ↑ | SSIM ↑ | PSNR ↑ | SSIM ↑ |
| RetinexNet (Wei et al., 2018) | 18.38 | 0.7756 | 19.92 | 0.8847 |
| KinD (Zhang et al., 2019) | 20.38 | 0.8248 | 22.62 | 0.9041 |
| ZeroDCE (Guo et al., 2020) | 16.8 | 0.5573 | 17.53 | 0.6072 |
| KinD++ (Zhang et al., 2021) | 21.3 | 0.8226 | 21.17 | 0.8814 |
| URetinex-Net (Wu et al., 2022) | 21.33 | 0.8348 | 22.89 | 0.895 |
| FECNet (Huang et al., 2022) | 22.24 | 0.8372 | 22.57 | 0.8938 |
| SNR-Aware (Xu et al., 2022) | 23.38 | 0.8441 | 24.12 | 0.9222 |
| FreqMamba (Zhen et al., 2024) | 23.57 | 0.8453 | 24.46 | 0.9355 |
| Ours | **23.78** | **0.8467** | **24.75** | **0.9452** |

*Table 10.* Comparison of methods on Dense-Haze and NH-HAZE datasets.

| Method | Dense-Haze | | NH-HAZE | |
|---|---|---|---|---|
| | PSNR ↑ | SSIM ↑ | PSNR ↑ | SSIM ↑ |
| DCP (He et al., 2010) | 10.06 | 0.3856 | 10.57 | 0.5196 |
| DehazeNet (Cai et al., 2016) | 13.84 | 0.4252 | 16.62 | 0.5238 |
| GridNet (Bozcan et al., 2021) | 13.31 | 0.3681 | 13.80 | 0.5370 |
| MSBDN (Dong et al., 2020) | 15.37 | 0.4858 | 19.23 | 0.7056 |
| AECR-Net (Wu et al., 2021) | 15.80 | 0.4660 | 19.88 | 0.7173 |
| FreqMamba (Zhen et al., 2024) | 17.35 | 0.5827 | 19.93 | 0.7372 |
| Ours | **18.91** | **0.6763** | **20.03** | **0.7508** |

operates in the Fourier space using only $1 \times 1$ convolutions, which fails to fully utilize the rich frequency information inherent to the Fourier domain. Specifically, FreqMamba applies Mamba scanning in a wavelet-transformed domain. However, the wavelet-transformed domain lacks the notable advantages of the Fourier domain, such as the Fourier transform's ability to decouple degradations and its global representation properties. Additionally, after wavelet decomposition, FreqMamba divides the image into multiple patches and performs spatial scanning within each patch. This design limits FreqMamba's ability to effectively model frequency correlations.

In contrast, our method performs Mamba scanning directly in the Fourier domain, fully leveraging the global characteristics of the Fourier transform. This allows our approach to better capture rain streaks, which often exhibit high apparent repetitiveness. Consequently, from a visual perspective, our method demonstrates significantly better performance in removing rain streaks. As shown in Figure 7, we show the feature maps and restoration results of FreqMamba and our method. It can be seen that our method can better capture the rain lines and thus remove the rain lines more cleanly.

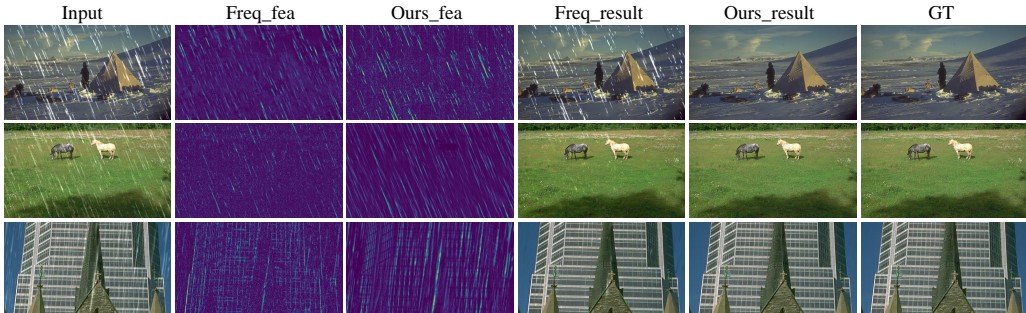

*Figure 7.* Feature maps and restoration results of FreqMamba and our method.

## A.10. Comparison with other methods such as DRSformer and FADformer

In this section, we compare our method with RCDNet (Wang et al., 2020),MPRNet (Zamir et al., 2021), SPDNet (Yi et al., 2021),DualGCN (Fu et al., 2021),HCN (Fu et al., 2023),Uformer (Wang et al., 2022b),IDT (Xiao et al., 2022),Restormer (Zamir et al., 2022),DRSformer (Chen et al., 2023a) and FADformer (Gao et al.), as shown in Table 11. To ensure fairness, we adopt the same experimental setup as the other methods, performing independent training and testing on each dataset, including Rain200L/H (Yang et al., 2017), DID-Data (Zhang & Patel, 2018), DDN-Data (Fu et al., 2017b), and SPA (Wang et al., 2019). The results demonstrate that our method achieves superior performance on the majority of the datasets.

Table 11. Performance comparison of methods across various datasets.

| Method | Rain200L | | Rain200H | | DID-Data | | DDN-Data | | SPA-Data | | Average | |
|---|---|---|---|---|---|---|---|---|---|---|---|---|
| | PSNR ↑ | SSIM ↑ | PSNR ↑ | SSIM ↑ | PSNR ↑ | SSIM ↑ | PSNR ↑ | SSIM ↑ | PSNR ↑ | SSIM ↑ | PSNR ↑ | SSIM ↑ |
| RCDNet | 39.17 | 0.9885 | 30.24 | 0.9048 | 34.08 | 0.9532 | 33.04 | 0.9472 | 43.36 | 0.9831 | 35.97 | 0.9554 |
| MPRNet | 39.47 | 0.9825 | 30.67 | 0.911 | 33.99 | 0.959 | 33.1 | 0.9347 | 43.64 | 0.9844 | 36.17 | 0.9543 |
| SPDNet | 40.5 | 0.9875 | 31.28 | 0.9207 | 34.57 | 0.956 | 33.15 | 0.9457 | 43.2 | 0.9871 | 36.54 | 0.9594 |
| DualGCN | 40.73 | 0.9886 | 31.15 | 0.9125 | 34.37 | 0.962 | 33.01 | 0.9489 | 44.18 | 0.9902 | 36.68 | 0.9604 |
| HCN | 41.31 | 0.9892 | 31.34 | 0.9248 | 34.7 | 0.9613 | 33.42 | 0.9512 | 45.03 | 0.9907 | 37.16 | 0.9634 |
| Uformer | 40.2 | 0.986 | 30.8 | 0.9105 | 35.02 | 0.9621 | 33.95 | 0.9545 | 46.13 | 0.9913 | 37.22 | 0.9609 |
| IDT | 40.74 | 0.9884 | 32.1 | 0.9344 | 34.89 | 0.9623 | 33.84 | 0.9549 | 47.35 | 0.993 | 37.78 | 0.9666 |
| Restormer | 40.99 | 0.989 | 32.0 | 0.9329 | 35.29 | 0.9641 | 34.20 | 0.9571 | 47.98 | 0.9921 | 38.09 | 0.9670 |
| DRSformer | 41.23 | 0.9894 | 32.17 | 0.9326 | 35.35 | 0.9646 | 34.35 | 0.9588 | 48.54 | 0.9924 | 38.32 | 0.9676 |
| FADformer | 41.80 | 0.9906 | 32.48 | 0.9359 | 35.48 | 0.9657 | 34.42 | **0.9602** | **49.21** | **0.9934** | 38.67 | 0.9691 |
| Ours | **42.27** | **0.9908** | **32.71** | **0.9395** | **35.49** | **0.9659** | **35.58** | 0.9599 | 49.18 | 0.9931 | **39.05** | **0.9698** |

## A.11. Difference between mamba and convolution in processing Fourier frequencies

First, Mamba utilizes sequence modeling to integrate information across all frequency bands, effectively leveraging the complementary relationships between different bands. In contrast, convolution, as a local operation, struggles to holistically model global features across all frequency bands when processing frequency information in the Fourier domain. This limitation significantly constrains its capacity in the Fourier space. Second, Mamba's sequence modeling is orderly, which can help the network establish an orderly dependency relationship between different frequencies. This characteristic is critical for modeling image degradation information. Conversely, convolution is insufficient in capturing the dependencies between high and low frequencies in the Fourier domain, thereby weakening its ability to accurately represent degradation features. In summary, based on these two advantages, Mamba achieves better coordination of high-frequency and low-frequency information in the Fourier domain during the image restoration process.

We process the Fourier frequencies using both Mamba and convolution separately, and then visualize their features, as shown in Figure 8. It can be seen our method (i.e., Mamba) not only captures rain streaks effectively but also extracts structural information from the background with high accuracy.

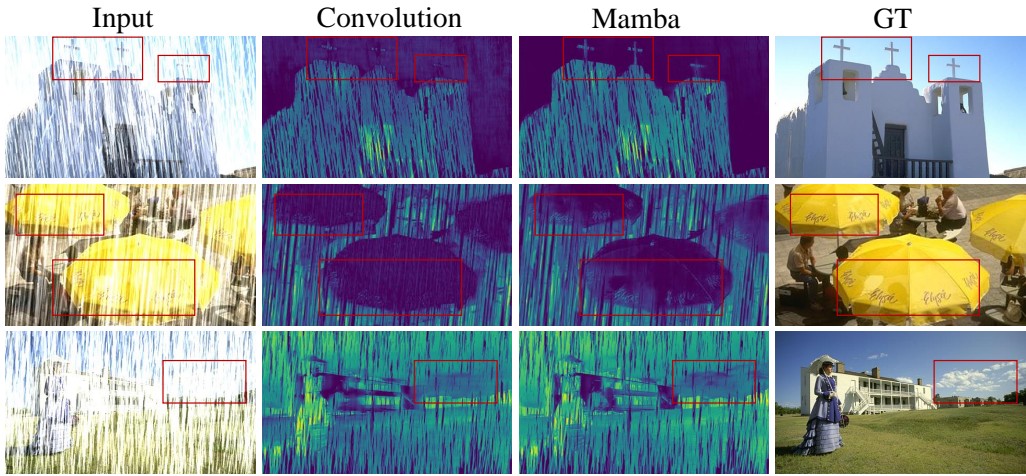

Figure 8. Feature visualization comparison of frequency-domain convolution and mamba on Rain100H.

## A.12. More visual deaining comparison on synthetic datasets

In this section, we provide more visual deraining comparisons on synthetic datasets to further demonstrate the effectiveness of our method. Specifically, we perform visual comparisons on several datasets in Table 1. Figure 9 shows more visualization results on Rain100H. As with the results in the main text, it shows that our method can better remove rain effects and prevent artifacts, which is attributed to the progressive frequency correlation. Figures 10, 11, and 12 show the visualization results

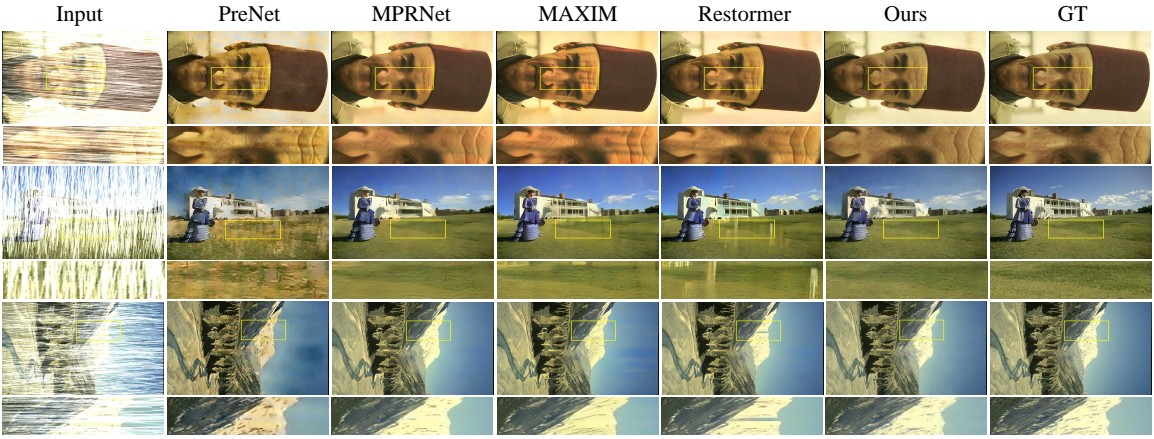

| Input | PreNet | MPRNet | MAXIM | Restormer | Ours | GT |

*Figure 9.* More qualitative comparison on Rain100H (Yang et al., 2017).

*Table 12.* Quantitative comparison of testing on the real-world dataset SPA-Data.

| Method | PReNet | RESCAN | HiNet | MSPFN | Restormer | MPRNet | Ours |
|--------|--------|--------|-------|-------|-----------|--------|------|
| PSNR | 31.33 | 31.56 | 33.89 | 34.03 | 34.18 | 34.54 | **35.27** |
| SSIM | 0.9501 | 0.9423 | 0.9500 | 0.9471 | 0.9493 | 0.9548 | **0.9575** |

on the simulation datasets Rain100L, Test2800, and Test1200, respectively.

### A.13. More real-world detraining results by using synthetic data

In this section, we provide more real-world rain removal cases to verify the generalization ability of the model trained on the synthetic dataset (rain13k). The quantitative comparisons directly tested on SPA-Data are shown in Table 12. The visualization results are shown in Figure 13. Our method is superior to other methods in rain removal and detail recovery. To further demonstrate its generalization ability in the real world, we also tested it on a real-world dataset RE-RAIN —(Chen et al., 2023b) , as shown in Figure 14. FourierMamba can obtain the most visually pleasing results. In addition, we also tested our method directly on RainDS-Real (Quan et al., 2021), and the quantitative results are shown in the table 13. Figure 15 shows the visualization results on RainDS-Real. It can be seen that our method can effectively remove real rain.

### A.14. More real-world visual deraining results by training real-world rainy images

Training and testing on real-world rainy images can verify the representation ability of the model in the real world. In Section 4.2, we use the real-world dataset SPA-Data to train FourierMamba and report quantitative results. In this section, we show the visualization results of training and testing using SPA-Data, as shown in Figure 16. It can be seen that our method excels at removing rain and recovering details to obtain pleasing visual results. In addition, we also train and test FourierMamba on RainDS-Real (Quan et al., 2021) to further verify its effectiveness in real-world scenes. As shown in Table 14, our method can still achieve excellent performance.

### A.15. Metrics that can better reflect human perceptions

In this section, we use some metrics that better reflect human perception to evaluate our method. We use the more widely used perceptual metrics BRISQUE (Mittal et al., 2012b), NIQE (Mittal et al., 2012a), SSEQ (Liu et al., 2014), as shown in Table 15. It can be seen that our method can also achieve excellent performance on perceptual metrics.

*Table 13.* Quantitative results of testing on the real-world dataset RainDS-Real (Quan et al., 2021).

| Method | PReNet | RESCAN | Restormer | HINet | MSPFN | MPRNet | Ours |
|--------|--------|--------|-----------|-------|-------|--------|------|
| PSNR | 24.15 | 24.29 | 24.54 | 24.71 | 24.76 | 25.07 | **25.12** |
| SSIM | 0.711 | 0.717 | 0.727 | 0.9731 | 0.729 | 0.736 | **0.738** |

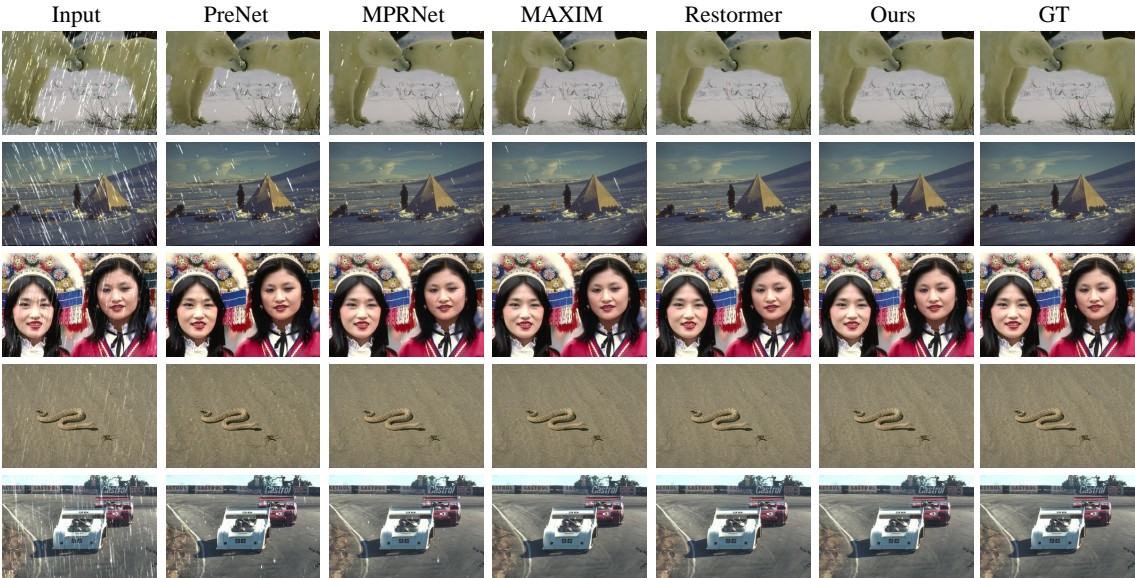

*Figure 10.* Qualitative comparison on Rain100L (Yang et al., 2017). Zoom in for better visualization.

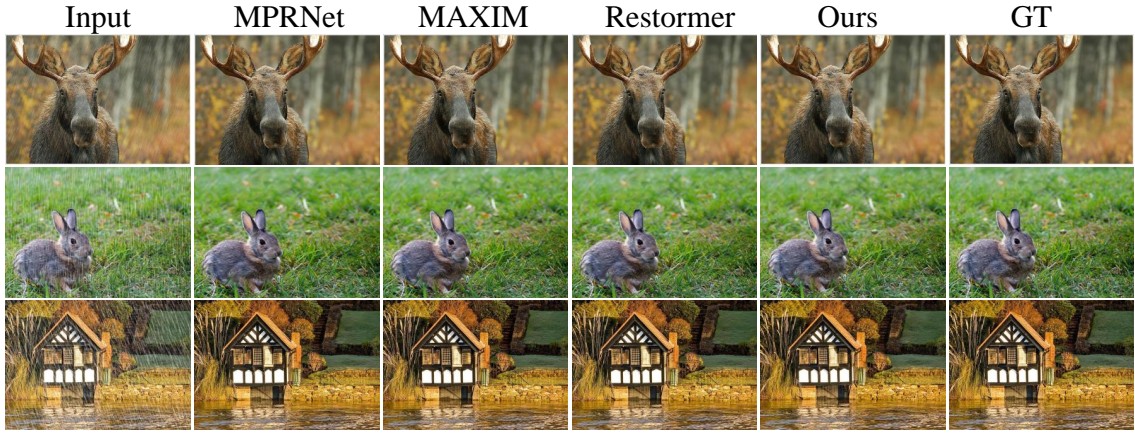

*Figure 11.* Qualitative comparison on Test2800 (Fu et al., 2017b). Zoom in for better visualization.

Input      PreNet      MPRNet      MAXIM      Restormer      Ours      GT

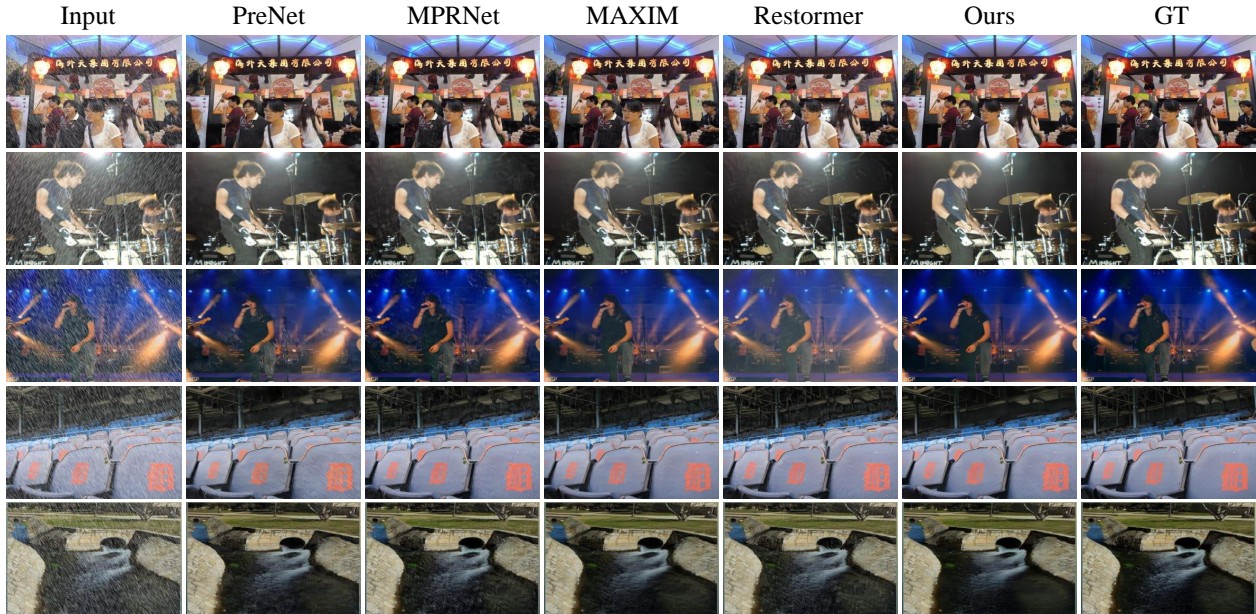

*Figure 12.* Qualitative comparison on Test1200 (Zhang & Patel, 2018). Zoom in for better visualization.

Input      PreNet      MPRNet      Restormer      Ours      GT

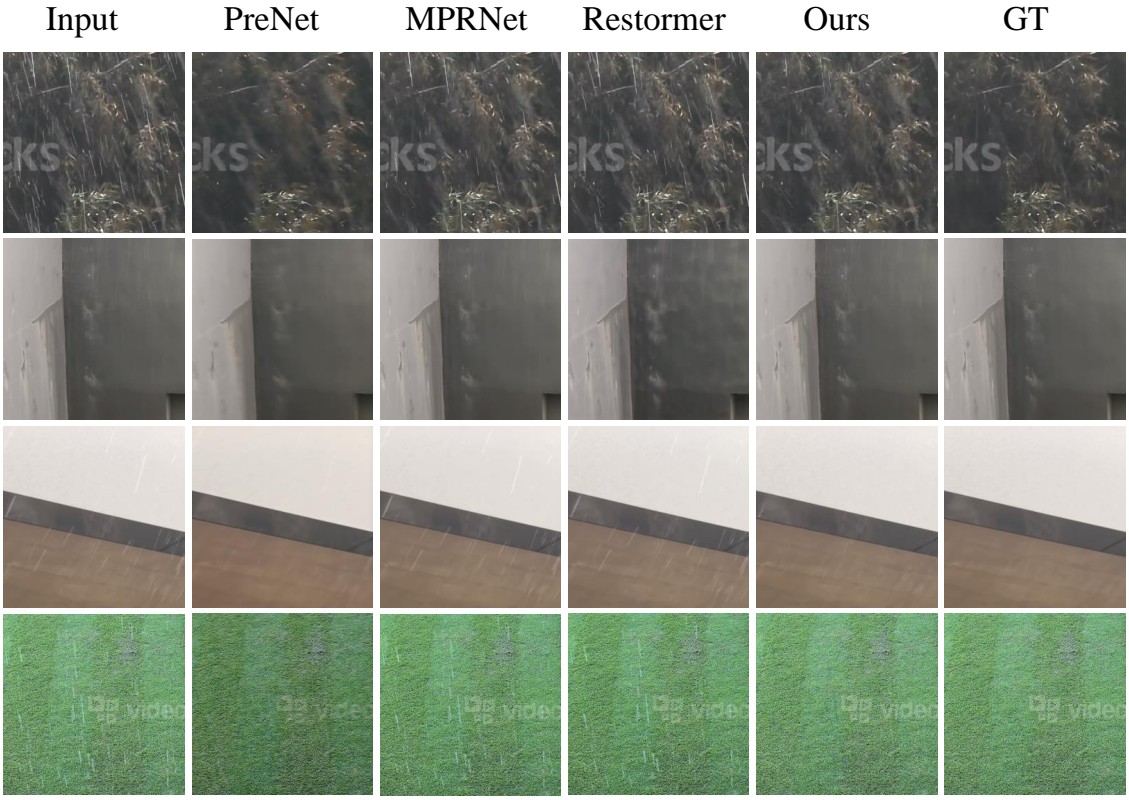

*Figure 13.* Qualitative comparison of real-world rainy images from SPA-Data(Wang et al., 2019).

Input      PreNet      MPRNet      Restormer      Ours

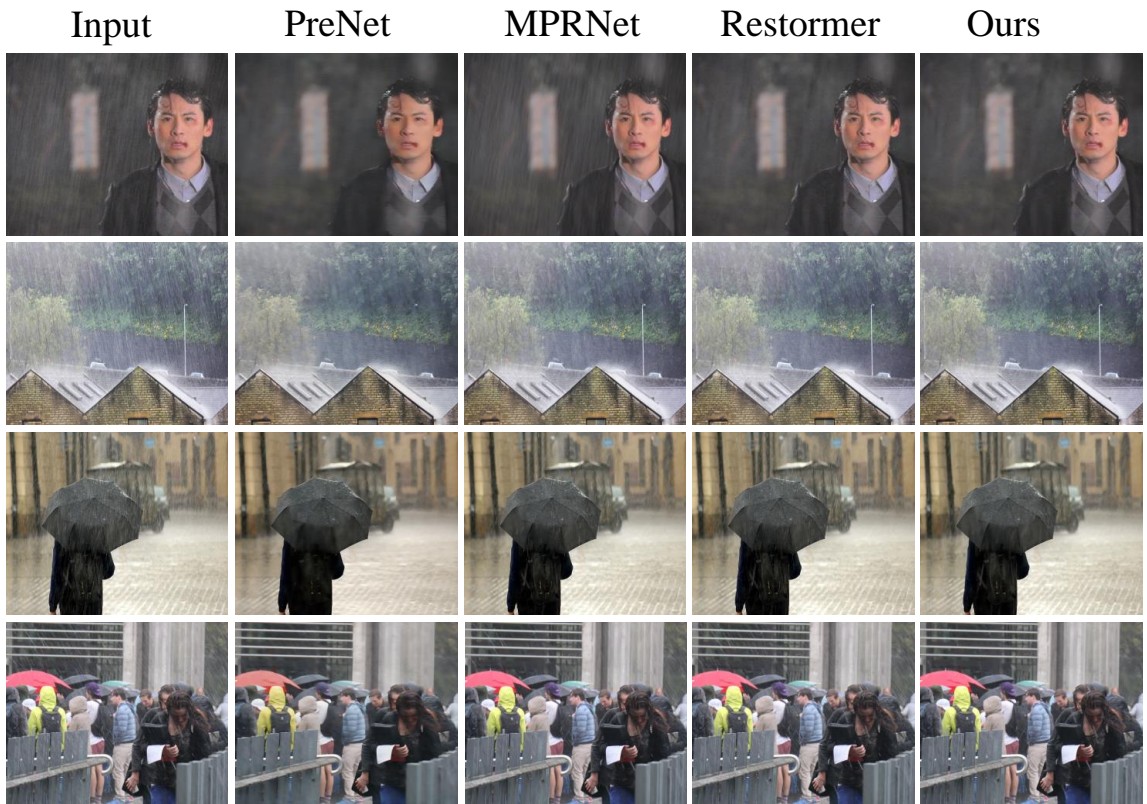

*Figure 14.* Qualitative comparison of real-world rainy images from RE-RAIN (Chen et al., 2023b).

Input    PreNet    MPRNet    Restormer    Ours    GT

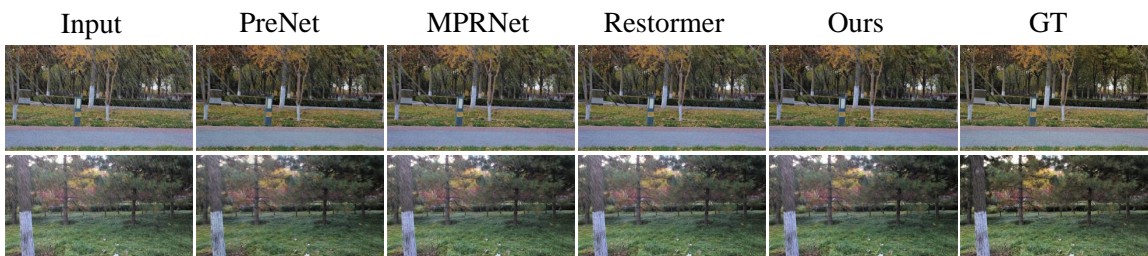

*Figure 15.* Qualitative results of real-world rainy images from RainDS-Real. (Quan et al., 2021).

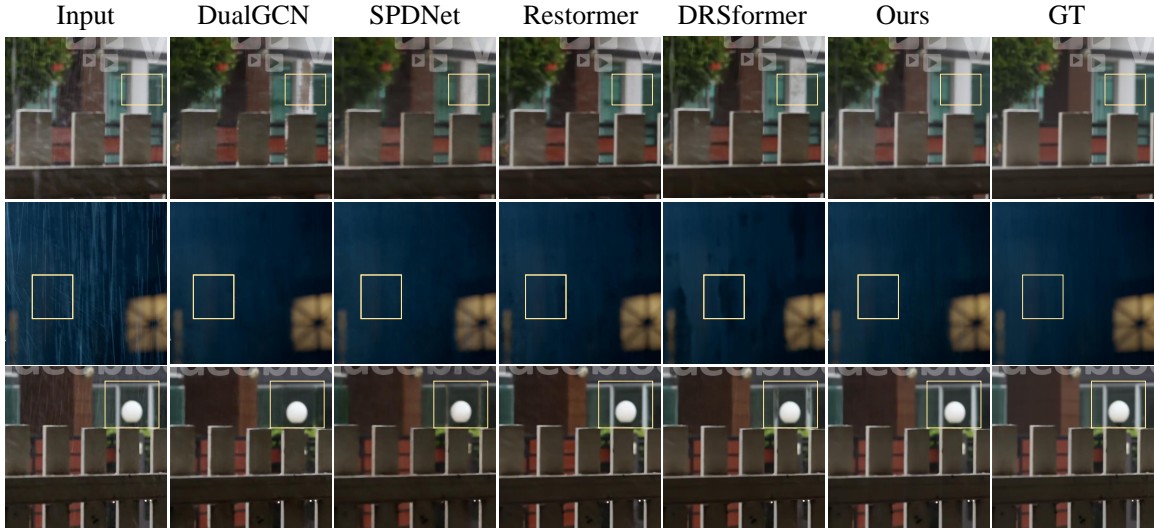

Input    DualGCN    SPDNet    Restormer    DRSformer    Ours    GT

*Figure 16.* Qualitative results of training and testing on SPA-Data. (Wang et al., 2019).

*Table 14.* Quantitative comparison of training and testing on the real-world dataset RainDS-Real.

| Method | PReNet | MSPFN | RCDNet | MPRNet | SwinIR | Restormer | Ours |
|--------|--------|-------|--------|--------|--------|-----------|------|
| PSNR | 26.43 | 26.45 | 26.71 | 27.51 | 27.53 | 27.57 | **27.69** |
| SSIM | 0.7294 | 0.7270 | 0.7180 | 0.7355 | 0.7425 | 0.7438 | **0.7482** |

## A.16. More about the Optimization

In the main body, we describe that apply the L1 loss based on the Fourier transform. Here, we introduce the loss function in the frequency domain in further detail. we first use the Fourier transform to convert $Y_{out}$ and $Y_{gt}$ into the Fourier space. Then, the $\mathcal{L}_1$-norm of the amplitude difference and phase difference between $Y_{out}$ and $Y_{gt}$ are calculated and summed to produce the total frequency loss as following:

$$\left\| \mathcal{F}(Y_{out}) - \mathcal{F}(Y_{gt}) \right\|_1 = \left\| \mathcal{A}(Y_{out}) - \mathcal{A}(Y_{gt}) \right\|_1 + \left\| \mathcal{P}(Y_{out}) - \mathcal{P}(Y_{gt}) \right\|_1. \tag{15}$$

## A.17. Ablation study on different frequency domain loss functions

We use three additional frequency domain loss functions: Phase Consistency Loss (PCL), Frequency Distribution Loss (PDL), and Focal Frequency Loss (FFL) (Jiang et al., 2021) for comparison with the L1 frequency domain loss we use. PCL is defined as the mean squared error of the phase difference between two images in the frequency domain, expressed as:

$$\mathcal{L}_{\text{PCL}} = \frac{1}{HW} \sum_{u,v} \left| \mathcal{P}(Y_{out})(u,v) - \mathcal{P}(Y_{gt})(u,v) \right|^2. \tag{16}$$

FDL represents the difference in frequency domain amplitude distributions between two images, expressed as:

$$\mathcal{L}_{\text{FDL}} = \frac{1}{HW} \sum_{u,v} \left| \mathcal{A}(Y_{out})(u,v) - \mathcal{A}(Y_{gt})(u,v) \right|^2. \tag{17}$$

FFL focuses on frequency components that are difficult to synthesize by down-weighting the easier ones, expressed as:

$$w(u,v) = |\mathcal{F}(Y_{out})(u,v) - \mathcal{F}(Y_{gt})(u,v)|^\alpha,$$
$$\mathcal{L}_{\text{FFL}} = \frac{1}{HW} \sum_{u=0}^{H-1} \sum_{v=0}^{W-1} w(u,v) |\mathcal{F}(Y_{out})(u,v) - \mathcal{F}(Y_{gt})(u,v)|^2, \tag{18}$$

where $\mathcal{F}(\cdot)(u,v)$ represents the Fourier Transform, $w(u,v)$ is the weight for the spatial frequency at $(u,v)$, and $\alpha$ is the scaling factor for flexibility ($\alpha = 1$ in practice).

*Table 15.* Performance comparison of different methods on various datasets. Metrics include BRISQUE, NIQE, and SSEQ.

| Dataset | Rain100L | | | Rain100H | | | Test2800 | | | Test1200 | | | Test100 | | |
|---|---|---|---|---|---|---|---|---|---|---|---|---|---|---|---|
| Method | BRISQUE ↓ | NIQE ↓ | SSEQ ↓ | BRISQUE ↓ | NIQE ↓ | SSEQ ↓ | BRISQUE ↓ | NIQE ↓ | SSEQ ↓ | BRISQUE ↓ | NIQE ↓ | SSEQ↓ | BRISQUE ↓ | NIQE ↓ | SSEQ ↓ |
| MPRNet | 17.791 | 6.816 | 12.702 | 16.287 | 6.973 | 13.860 | 15.782 | 6.251 | 9.470 | 23.434 | 5.742 | 12.653 | 23.526 | 6.903 | 12.767 |
| MAXIM | **11.960** | 6.402 | 9.658 | 14.622 | 6.929 | 8.034 | 15.272 | 6.114 | 8.760 | 25.026 | 5.573 | 14.760 | 22.433 | 6.770 | 12.615 |
| Restormer | 16.253 | 6.555 | 11.480 | 17.606 | 6.843 | 13.953 | 18.601 | 6.169 | 9.579 | 25.507 | **5.534** | 16.121 | 23.937 | 7.024 | 14.382 |
| MambaIR | 15.662 | 6.553 | 11.527 | **10.350** | 6.104 | 8.719 | 13.246 | 6.165 | 8.332 | **20.743** | 5.570 | 10.877 | 17.886 | 5.969 | 8.390 |
| VmambaIR | 16.073 | 6.651 | 11.061 | 11.686 | 5.713 | 8.302 | 13.465 | 6.114 | 8.306 | 20.851 | 5.553 | **10.610** | 17.805 | **5.751** | 8.548 |
| FreqMamba | 14.894 | 6.465 | 10.286 | 15.151 | **5.450** | **4.704** | 19.942 | 5.439 | 10.371 | 22.132 | 5.785 | 10.742 | 18.934 | 5.898 | 7.247 |
| Ours | 12.178 | **6.261** | 10.008 | 10.607 | 6.009 | 5.826 | **12.895** | **5.258** | 8.286 | 21.467 | 5.538 | 10.839 | **17.738** | 5.958 | **7.102** |

We conduct ablation comparison experiments on these loss functions, as shown in Table 16. It can be seen that the performance obtained by these four loss functions is similar. The focus of this work is on the design of the network architecture, so we follow existing methods (Zhou et al., 2023; Zhen et al., 2024) to use the L1 norm in the frequency domain. We will explore more frequency-domain loss functions in future work.

*Table 16.* Comparison results of different frequency-domain loss functions.

| | PCL | FDL | FFL | Ours |
|---|---|---|---|---|
| PSNR | 39.67 | 39.69 | 39.75 | 39.73 |
| SSIM | 0.9848 | 0.9852 | 0.9859 | 0.9856 |

## A.18. More visualizations for ablation studies

In Section 4.3, we perform ablation studies on the key designs and scanning methods of the proposed method. To further verify the effectiveness of the proposed method, we provide visualizations of the above ablation studies. Specifically, we subtract the restored images obtained from each ablation study from the ground truth to show the effect of each design. Figure 17 shows the visualization of the ablation study in Table 3. It can be seen that all designs have a significant effect on rain removal. Figure 18 shows the visualization of the ablation study of various scanning methods in Table 4. Both Figures 4 and Figures 18 illustrate that orderly correlation of different frequencies can promote rain removal.

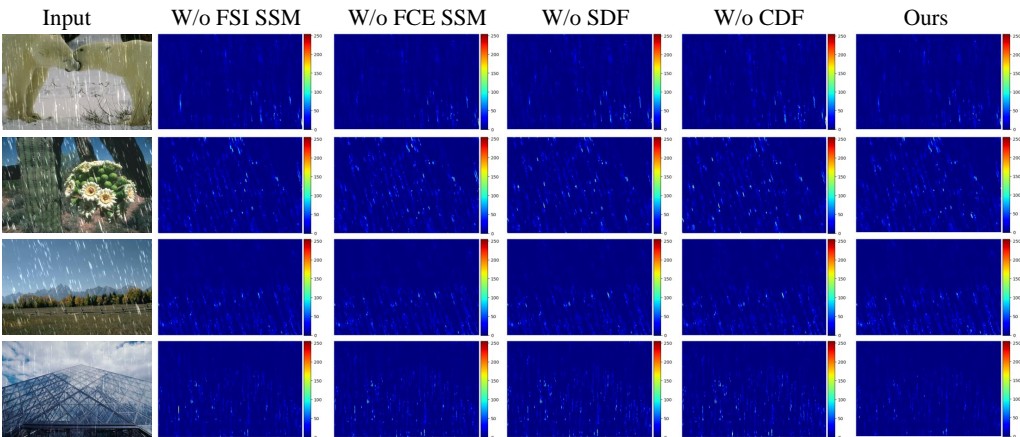

*Figure 17.* Visualization of ablation studies of various key designs of the proposed method.

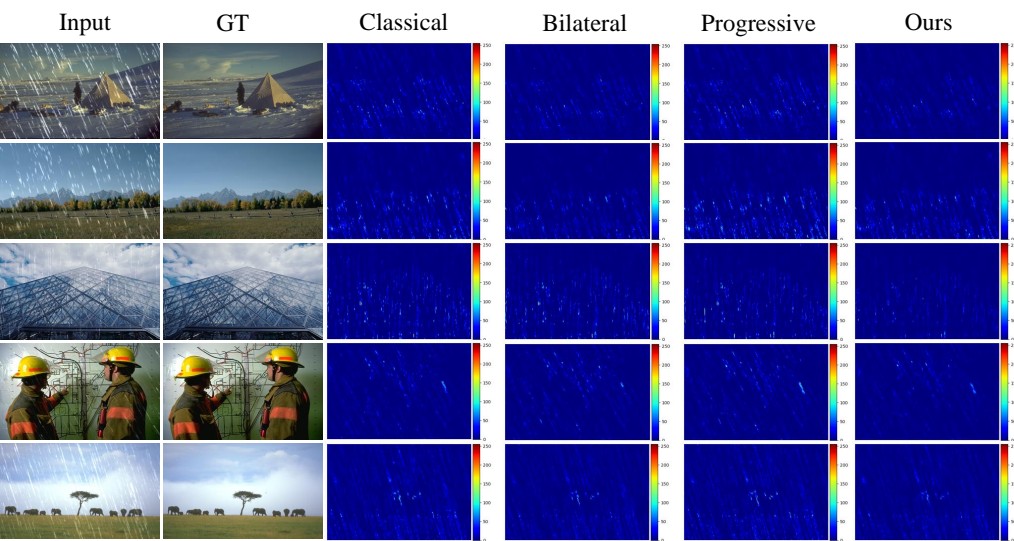

*Figure 18.* Visualization of ablation studies of different scanning methods in Fourier space.

