# OpenReview forum: "FourierMamba: Fourier Learning Integration with State Space Models for Image Deraining"
_ICML.cc/2025/Conference — ICML 2025 poster_

### Official Review · Reviewer_Rwtp · 2025-03-12

**Overall Recommendation:** 4

**Summary:**

The authors propose a novel framework, FourierMamba, which integrates Fourier priors with a state-space model to associate different frequencies in the Fourier domain for image deraining.

## update after rebuttal
The authors' second-round response has addressed most of my concerns. I now find the motivation of the paper to be reasonable and the experiments to be sufficiently thorough. I have decided to raise my score.

**Claims And Evidence:**

The motivation behind combining Mamba with the Fourier space needs to be discussed in greater depth. The results shown in Figure 1 only include 1×1 convolution and previous scanning in Fourier space. However, more commonly used alternatives, such as Transformers and stacked 3×3 convolutions, should also be considered and discussed.

**Essential References Not Discussed:**

n/a

**Experimental Designs Or Analyses:**

If the authors provide more real-world examples and analyze the contributions of Fourier and Mamba in these scenarios, this work would  more convincing.

**Methods And Evaluation Criteria:**

The authors lack quantitative evaluation in real-world scenarios, yet the ultimate goal of image processing is to apply it to real-world situations. Quantitative assessment of real rain images without ground truth (GT), using no-reference evaluation metrics, is crucial for demonstrating the method's practical applicability.

**Other Comments Or Suggestions:**

As mentioned earlier, real-world evaluation is a crucial aspect of validating image deraining methods. However, Figure 6 is too small, making it difficult to discern the differences between different methods. The authors may consider removing some of the early baseline methods to improve clarity and focus on more recent approaches.

**Other Strengths And Weaknesses:**

Strengths:
1.The authors combine Mamba with the Fourier space, which is an interesting idea.
2. The proposed method achieves state-of-the-art performance on multiple image deraining datasets.
3. This work is evaluated on multiple image restoration tasks beyond image deraining, further demonstrating the model's generalization capability and effectiveness.

Weaknesses:
1.The Mamba network suffers from two issues: local pixel forgetting and spatial misalignment. I am curious about how these issues manifest in the Fourier domain. Would they still occur, or do the Fourier priors help mitigate these problems?
2.The equations in the Preliminary section are not clearly linked to the modules in Figure 3. In other words, these equations are not directly utilized later in the paper, making them appear redundant. The module primarily applies FFT for phase and amplitude separation, which has a weak connection to the equations presented in the Preliminary section.
3.The authors should include more examples from real-world scenarios to further demonstrate the model's generalization ability.
4.Some references need to be updated, changing citations from arXiv papers to the officially accepted versions of the papers.
5.The double-line and triple-line tables in the manuscript should be standardized for consistency.
6.In Figure 7, I noticed that besides the rain streaks, the feature maps also contain window edge textures. Could this lead to an issue where the Fourier space mistakenly treats window edges as having similar frequency characteristics to rain streaks, potentially causing background degradation?
7.I am curious why many entries in Table 1 are missing. As far as I know, these methods have publicly available source codes. Could the authors clarify why the results for these methods are not reported?
8.The font size in tables and figures is inconsistent (e.g., Tables 2, 3, and 4). The authors should standardize the font size across all tables and figures for consistency.

**Questions For Authors:**

n/a

**Relation To Broader Scientific Literature:**

n/a

**Theoretical Claims:**

n/a

---

> ### Author Rebuttal · Authors · 2025-04-01
>
> **R1**: Clarification on Motivation
>
> Motivation: Mamba, a state-space model, offers global modeling and linear complexity, making it ideal for sequential data. Frequency information in the Fourier domain is inherently global, and Mamba’s sequence modeling efficiently captures inter-frequency dependencies. In contrast, Transformers incur high computational complexity (O(n²)) for long sequences, while 3×3 convolutions are limited by their receptive fields in capturing global frequency data. Thus, integrating Mamba with the Fourier domain is a targeted signal-processing design, leveraging frequency globality and computational efficiency to enhance deraining performance innovatively.
> Comparative Experiments: Addressing the suggestion, we conducted experiments comparing Transformer and 3×3 convolution on the Rain200L dataset (Table 1). Results demonstrate that Mamba’s scanning approach effectively balances performance and efficiency.
>
> **R2**: Response to “Lack of Quantitative Evaluation in Real-World Scenarios”
>
> We have included no-reference quantitative evaluation results for the Internet test set without ground truth (GT) from the SPA dataset, as presented in Table 2.
>
> **R3**: Response to “Experimental Design and Analysis”
>
> Real-world deraining results are presented in Figures 13–16, with quantitative outcomes in Tables 13–15, demonstrating adaptability and generalizability to real rain due to Fourier priors and Mamba’s efficient global modeling.
>
> **R4**: Response to “Issues with Mamba in the Fourier Domain”
>
> Mamba’s local pixel forgetting in the spatial domain stems from its sequence modeling, potentially overlooking local details, while spatial misalignment relates to scanning order. In the Fourier domain, however, information is globally represented as frequencies, rendering local forgetting less prominent, as frequency components reflect overall image properties. Moreover, Fourier priors decompose the image into frequencies, aiding Mamba in modeling global information comprehensively and mitigating misalignment. Experimental results (Figures 5 and 9) show FourierMamba preserves background details and removes rain streaks effectively, confirming these issues are alleviated in the Fourier domain. This analysis will be added to the revision.
>
> **R5**: Response to “Unclear Connection Between Preliminary Equations and Figure 3 Modules”
>
> We acknowledge the lack of clarity and will refine the Preliminary section in the revised manuscript. The equations therein introduce Fourier transform fundamentals, laying the theoretical groundwork for subsequent frequency modeling. The FFT operation in Figure 3 directly applies these principles, converting images from the spatial to the Fourier domain and separating amplitude and phase. We will explicitly link these equations to the FFT module in Figure 3, enhancing logical coherence and eliminating perceived redundancy.
>
> **R6**: Response to “Issues with Feature Maps in Figure 7”
>
> Figure 7 compares feature maps of FreqMamba and our method, revealing that our approach more effectively focuses on and localizes rain streak-related features. Consequently, our derain results (Fig5-6, Fig9-16) retains background information while minimizing residual rain streaks compared to FreqMamba.
>
> **R7**: Response to “Missing Entries in Table 1”
>
> Table 1 references the table in FreqMamba. Missing data primarily result from unavailable open-source code or discrepancies between open-source implementations and the corresponding papers, leading us to directly adopt reported results from the papers. In the future, we will strive to complete missing entries or provide explanations as feasible.
>
> **R8**: Response to “Formatting and Presentation Issues”
>
> In the revised manuscript, we will adjust the font sizes of tables and figures for consistency.
>
> **R9**: Response to “Layout Issues in Figure 6”
>
> Our method’s advantages over others in real-world scenarios are observable through enlarged patches. We will revise the layout of Figure 6 in the updated manuscript. Additional visual results for real-world deraining can be found in Supplementary Figures 13–16.
>
>
>
> | Method      | PSNR  | SSIM   | Flops | Params |
> |-------------|-------|--------|-------|--------|
> | 3*3 Conv    | 40.24 | 0.9887 | 17.44 | 21.98  |
> | Transformer | 42.21 | 0.9896 | 18.98 | 72.49  |
> | Ours        | 42.27 | 0.9908 | 17.62 | 22.56  |
>
> | Method      | BRIQUE ↓ | NIQE ↓ | SSEQ ↓ |
> |-------------|----------|--------|--------|
> | Rainy Input | 28.517   | 5.095  | 28.280 |
> | MPRNet      | 34.733   | 5.144  | 33.765 |
> | Restormer   | 32.288   | 4.851  | 31.789 |
> | IDT         | 27.042   | 4.536  | 28.314 |
> | DRSformer   | 26.080   | 4.531  | 27.954 |
> | FADformer   | 25.959   | 4.760  | 26.667 |
> | Freqmamba   | 26.172   | 4.890  | 27.387 |
> | Ours        | 25.827   | 4.682  | 26.423 |

---

> > ### Comment · Reviewer_Rwtp · 2025-04-02
> >
> > About R1 and R4: The authors mention that in the Fourier domain, information is globally represented in the form of frequency components, which makes local forgetting less prominent. I would like to ask: Does this global representation in the Fourier domain diminish the advantage of Mamba’s global receptive field ? Additionally, since convolution in the Fourier domain can also capture global information, what distinct benefit does Mamba offer in this setting compared to simpler operations like Fourier-based convolution?
> > Clarifying these points would help better understand the specific role and necessity of Mamba within the Fourier domain.

---

> > > ### Author Response · Authors · 2025-04-02
> > >
> > > **Response to Follow-Up Questions on R1 and R4**
> > >
> > > We appreciate the reviewer’s further inquiries on R1 and R4, which allow us to clarify Mamba’s specific role and necessity in FourierMamba. Below, we address the two questions in detail.
> > >
> > > **Question 1**: Does the Global Representation in the Fourier Domain Diminish the Advantage of Mamba’s Global Receptive Field?
> > >
> > > The Fourier transform decouples an image into distinct frequency bands, with each band aggregating global information from the entire spatial domain—its globality lies in information aggregation, where each frequency component integrates contributions from all spatial positions. In contrast, the proposed Mamba excels at uncovering relationships among these decoupled bands, with its globality manifested in cross-band sequential dependency modeling, capturing interactions between high frequencies (e.g., rain streaks) and low frequencies (e.g., background structures).
> > >
> > > Their combination leverages complementary strengths: the Fourier transform provides a global frequency decomposition, while Mamba’s global receptive field models inter-band dependencies. This synergy enhances comprehension of frequency interactions, crucial for tasks like image deraining, where separating degradation from clean content in the frequency domain requires accurately capturing their relationships.
> > >
> > > **Question 2**: What Distinct Benefit Does Mamba Offer in the Fourier Domain Compared to Convolution Operations?
> > >
> > > Mamba’s distinct advantage lies in its global sequence modeling capability, which captures dynamic dependencies between frequency bands, excelling notably in deraining tasks. Rain streak features, due to their complex variability in brightness, width, and length, exhibit significant diversity in pixel values and spatial scales. For instance, fine, short rain streaks may appear as high-frequency components, whereas coarse, long, and brighter streaks may span both low and high frequencies. This variability renders rain streaks difficult to fully represent with a single frequency band, necessitating the integration of information across multiple bands in the frequency domain.
> > >
> > > Compared to convolution operations in the Fourier domain, which are inherently local and limited to capturing relationships between adjacent frequency bands, convolution struggles to model long-range frequency dependencies due to its restricted receptive field. In contrast, Mamba’s global receptive field enables it to transcend frequency band boundaries, directly capturing long-range dependencies. For example, in deraining, Mamba correlates high-frequency rain streak features with low-frequency background information, effectively identifying and separating these complex cross-band patterns. Thus, while the Fourier transform provides the foundation of global frequency information, Mamba enhances the understanding of dynamic inter-band interactions, significantly improving the modeling of rain streak features in deraining tasks.
> > >
> > > **Summary**
> > >
> > > In the Fourier domain, convolution is confined by its locality to modeling adjacent frequency relationships, whereas Mamba’s globality effectively captures cross-band dependencies. This capability provides Mamba a distinct advantage in tasks requiring global frequency information, such as enhanced identification and separation of cross-band features in images. Comparative experiments from prior responses, alongside Table 3 in the main text and Figure 8 in the appendix, strongly support this analysis.

---

### Official Review · Reviewer_X9L3 · 2025-03-12

**Overall Recommendation:** 3

**Summary:**

The paper applies Mamba to dual-domain spatial dimensions and channels for image deraining. For spatial-dimensional Fourier processing, the authors introduce a new scanning method that better models the correlations between different frequencies. The constructed network is evaluated on multiple image restoration tasks beyond deraining and achieves promising performance.

**Claims And Evidence:**

The quantitative and qualitative results demonstrate the effectiveness of the proposed method.

**Essential References Not Discussed:**

N/A

**Experimental Designs Or Analyses:**

This paper closely follows the FreqMamba (MM'24) method. However, there is an inconsistency between these two methods regarding using datasets. Specifically, it seems that FreqMamba trains separate models for each deraining dataset while the proposed method is trained on a mixed dataset, Rain13k. The authors also mention this issue in the paper. Will this inconsistency impact the performance?

**Methods And Evaluation Criteria:**

The authors do experiments on standard datasets and metrics. In addition, the evaluation is performed using non-reference metrics.

**Other Comments Or Suggestions:**

n/a

**Other Strengths And Weaknesses:**

The proposed method introduces a new scanning method for image deraining to model correlations between Fourier frequencies. State-of-the-art performance is achieved on some image restoration tasks.

The reviewer's concerns are as follows:

1. The proposed network mainly follows that of FreMamba. The novelty of the method only lies in using a different scanning method for spatial-dimensional frequency features. Using Mamba to scan the channel features is direct and not novel in image restoration. Overall, the novelty is a little bit limited.

2. The experiments are not convincing, as the proposed method employs a different training strategy for the dataset compared to FreqMamba.

3. On some additional datasets, the performance is not competitive. For example, on SPA-Data, AST achieves 49.51 dB, which is much higher than the proposed method. For low-light enhancement, the proposed method is inferior to MambaLLIE while using more parameters.

4. The authors provide the speed comparisons in the supplementary material. The reviewer finds that the proposed method does not run fast in the deraining domain.

[AST] Adapt or Perish: Adaptive Sparse Transformer with Attentive Feature Refinement for Image Restoration, CVPR24.

[MambaLLIE] MambaLLIE: Implicit Retinex-Aware Low Light Enhancement with Global-then-Local State Space, NeurIPS24.

**Questions For Authors:**

Please refer to the entries above.

**Relation To Broader Scientific Literature:**

This paper introduces an image deraining architecture incorporating the zigzag scanning method with Mamba. The network achieves promising performance on some image restoration datasets.

**Theoretical Claims:**

In addition to the foundational equations for the Fourier transform, no other theoretical claims are provided.

---

> ### Author Rebuttal · Authors · 2025-04-01
>
> **R1**: Response to “Insufficient Novelty”
>
> The distinctions between our method and FreqMamba are detailed in Appendix A.9.
>
> Here, we further clarify the novelty:
>
> **Spatial-Domain Zigzag Scanning**: Unlike FreqMamba’s simple wavelet-domain scanning, FourierMamba’s zigzag scanning, inspired by JPEG encoding, orders low-to-high frequencies in the Fourier domain, enhancing correlation modeling while preserving symmetry. This integrates signal processing knowledge into Mamba’s efficient structure, synergistically improving deraining outcomes.
>
> **Channel-Domain Fourier Scanning**: Discussed in Appendix A.6, this design leverages varying degradation characteristics across channels, which collectively define global image information. By introducing Fourier transforms in the channel dimension, we capture inter-channel frequency dependencies, enhancing global representation while decoupling degradation (e.g., rain streaks) from background content using amplitude and phase spectra. Joint modeling with Mamba further strengthens channel interactions and global information modeling.
>
> **Overall Novelty**: FourierMamba combines spatial zigzag scanning and channel Mamba scanning to form a versatile deraining framework, outperforming MambaIR and FreqMamba. This dual-domain frequency modeling represents a significant advancement in image restoration, far beyond a mere scanning improvement.
>
> **R2**: Response to “Experiments Lack Convincing Evidence”
>
> Joint training on Rain13k demands greater network generalization, often yielding lower performance than models trained separately on individual datasets. Except for FreqMamba, results in Table 1 reflect joint training, whereas FreqMamba’s paper conflates the two approaches. For a fair comparison with FreqMamba, we provide results under identical experimental settings in Appendix Tables 11 and 16, showing improvements of 0.54 dB and 0.71 dB on Rain13k and SPA datasets, respectively, with our method.
>
> **R3**: Response to “Performance Lacking Competitiveness on Certain Datasets”
>
> AST was trained on the enhanced SPAD version[1] of the SPA dataset, making direct comparison unfair. Thus, we retrained AST on the SPA dataset using parameters from its paper, yielding the following results in Tab1. For low-light enhancement, FourierMamba targets image deraining, with experiments in this task only validating its generalizability, lacking specialized designs like MambaLLIE’s Retinex prior. Hence, it is reasonable that FourierMamba underperforms MambaLLIE in low-light enhancement. Future work will explore tailoring FourierMamba for this task.
>
> **R4**: Response to “Suboptimal Inference Speed”
> It was noted that FourierMamba’s inference time lags behind Restormer’s. This stems from PyTorch’s extensive CUDA optimizations for attention-based operators, which Mamba currently lacks. However, our method’s FLOPS (22.5) is significantly lower than Restormer’s (174.7), suggesting that further CUDA optimization of Mamba operators will enhance inference speed. Meanwhile, compared to other Mamba-based methods, ours achieves a superior balance of inference efficiency and performance.
>
> **R5**: Response to “Issues with Supplementary Material”
>
> The review highlighted a missing table reference in Section A.10 and an incorrect citation for FADformer. We apologize for these oversights and will correct them in the revised manuscript: Section A.10 will explicitly reference Table 12, and the FADformer citation will be updated to the correct source. We appreciate your meticulous feedback.
>
> | Method | PSNR  | SSIM   |
> |--------|-------|--------|
> | AST    | 48.49 | 0.9924 |
> | Ours   | 49.18 | 0.9931 |
>
> [1] Learning Weather-General and Weather-Specific Features for Image Restoration Under Multiple Adverse Weather Conditions [CVPR 2023]

---

> > ### Comment · Reviewer_X9L3 · 2025-04-09
> >
> > Thank the authors for the detailed response. My concerns have been addressed, and I have increased my rating accordingly. Congratulations.

---

### Official Review · Reviewer_q47J · 2025-03-13

**Overall Recommendation:** 4

**Summary:**

This paper proposes FourierMamba that addresses the problem of single image deraining by introducing a scanning encoding mechanism that correlates different frequencies in both spatial and channel dimensions. Specifically, it employs zigzag coding in the spatial dimension to reorganize frequency orders and improve their connectivity, while utilizing tailored designs in the channel dimension. This approach enables more effective frequency information utilization, leading to improved image deraining performance.

**Claims And Evidence:**

Yes.

**Essential References Not Discussed:**

N/A.

**Experimental Designs Or Analyses:**

The experimental analyses are sound because the authors have conducted extensive experiments on five datasets and shown the superiority of their method.

**Methods And Evaluation Criteria:**

Yes.

**Other Comments Or Suggestions:**

N/A

**Other Strengths And Weaknesses:**

Strengths:

1. The authors have conducted extensive experiments on over five commonly used deraining datasets, demonstrating the proposed method's superiority in both complexity and deraining performance, evaluated quantitatively and qualitatively.

2. The incorporation of zigzag coding in Fourier space and the concept of scanning encoding for different frequencies introduce a novel approach to deraining.

3. The paper is well-written and easy to follow, with numerous visual comparisons and feature visualizations effectively illustrating the deraining results of the proposed method.

Weaknesses:

1. To improve the implementation and evaluation parts, it is helpful to provide more details on the zigzag scanning implementation and its impact on inference speed.

2. To expanded comparisons and additional experiments, it is suggested to discuss the comparisons and differences between the proposed method and FreqMamba in Fourier correlation strategies. It is helpful to incorporate alternative perceptual metrics beyond PSNR and SSIM to align with human perception on the real-world rainy dataset such as SPA-Data.

3. It is recommended to justify why Mamba is more effective than other architectures for processing Fourier frequencies. The authors are also suggested to investigate generalization to other tasks (e.g., deblurring) and discuss the applicability of wavelet transformation.

**Questions For Authors:**

Please refer to the weaknesses.

**Relation To Broader Scientific Literature:**

The related works section overlooks several recent studies, including [1-2], which explore the concepts of pixel-level alignment and the linear model.

[1] Cross-Modality Fusion Mamba for All-in-One Extreme Weather-Degraded Image Restoration, 2025.
[2] Restoring images in adverse weather conditions via histogram transformer, 2024.

**Theoretical Claims:**

N/A, there is no theoretical claims in the paper.

---

> ### Author Rebuttal · Authors · 2025-04-01
>
> **R1**: Details of Zigzag Scanning Implementation and Its Impact on Inference Speed
>
> Section 3.2 briefly outlines the motivation and design of zigzag scanning, inspired by JPEG’s zigzag encoding, to order frequencies in the Fourier domain from low to high. For the spatial Fourier spectrum, it is divided into symmetric halves, with zigzag path scanning applied to one half, arranging frequencies from center (low) to edges (high), followed by Mamba sequence modeling. The other half is derived via Fourier symmetry, ensuring orderly frequency correlation without compromising symmetry, unlike full-spectrum scanning.
>
> As a preprocessing step, zigzag scanning’s computational cost, mainly from frequency rearrangement and Mamba modeling, is minimal. Experiments show its additional time is negligible, involving a one-time index reorder—reusable as a dictionary for same-resolution images—while Mamba’s linear complexity ensures efficiency. FourierMamba’s inference time matches MambaIR and FreqMamba, indicating no significant burden. Revised Section 3.2 will detail the implementation and confirm its minimal impact on inference speed.
>
> **R2**: Comparison with FreqMamba in Frequency-Domain Correlation Strategies
>
> We further clarify the differences and similarities between FourierMamba and FreqMamba in frequency-domain strategies. While FreqMamba employs Mamba in the frequency domain, it primarily conducts spatial scanning in the wavelet domain, underutilizing the global properties of the Fourier domain. In contrast, FourierMamba applies Mamba directly in the Fourier domain, using zigzag encoding for orderly frequency correlation, enhancing rain streak capture. FreqMamba’s wavelet-domain scanning limits its global frequency modeling capacity.
>
> Quantitative comparisons with FreqMamba on Rain13k and SPA datasets, shown in Tables 11 and 16, reveal our method’s improvements of 0.54 dB and 0.71 dB, respectively. Feature map visualizations in Figure 7 demonstrate that our approach more effectively targets rain degradation, yielding superior deraining results.
>
> **R3**: Diversity of assessment indicators
>
> We have augmented the evaluation with no-reference metric results for the unpaired Internet-Data test set from the SPA dataset, as shown in Table 1 above.
>
> **R4**: Effectiveness and Generalizability of Mamba in Processing Fourier Frequencies
>
> Mamba, a state-space model, excels with global modeling and linear complexity, ideal for sequential data. In the Fourier domain, where frequency information is inherently global and sequential, Mamba efficiently captures inter-frequency dependencies. Traditional convolutional or Transformer architectures, however, are less efficient or computationally costly for global frequency processing, while Mamba’s linear global attention effectively combines their strengths.
>
> Regarding generalizability, supplementary results demonstrate FourierMamba’s strong performance in low-light enhancement and dehazing on datasets like LOL-V1 and Dense-Haze, indicating robust task adaptability. For deblurring, where frequency loss is key, its correlation modeling shows promise, to be explored further in future work.
>
> Compared to wavelet transforms, which excel in local frequency and multi-scale analysis, Fourier transforms better support global frequency representation and degradation decoupling, particularly for deraining, where rain streaks are distinctly separable in the frequency domain. Thus, we prioritize Fourier transforms, but future work will explore wavelet integration. These points will be detailed in Sections 2 and 5 of the revised manuscript to fully address Mamba’s effectiveness and the method’s generalizability.
>
> **R5**: Related Works Omissions
>
> We appreciate your feedback and will include discussions of [1] and [2] in the revised manuscript. While [1] explores cross-modal fusion with Mamba for image restoration, our work focuses on frequency-domain modeling. Similarly, [2] employs linear models for weather degradation, which aligns partially with our approach. We will clarify these connections in Section 2 to better position our contributions.
>
> | Method      | BRIQUE ↓ | NIQE ↓ | SSEQ ↓ |
> |-------------|----------|--------|--------|
> | Rainy Input | 28.517   | 5.095  | 28.280 |
> | MPRNet      | 34.733   | 5.144  | 33.765 |
> | Restormer   | 32.288   | 4.851  | 31.789 |
> | IDT         | 27.042   | 4.536  | 28.314 |
> | DRSformer   | 26.080   | 4.531  | 27.954 |
> | FADformer   | 25.959   | 4.760  | 26.667 |
> | Freqmamba   | 26.172   | 4.890  | 27.387 |
> | Ours        | 25.827   | 4.682  | 26.423 |
>
> [1] Cross-Modality Fusion Mamba for All-in-One Extreme Weather-Degraded Image Restoration, 2025. [2] Restoring images in adverse weather conditions via histogram transformer, 2024.

---

### Official Review · Reviewer_qXRb · 2025-03-15

**Overall Recommendation:** 5

**Summary:**

This paper introduces FourierMamba, a novel framework for image deraining that leverages the Mamba technique within the Fourier space to effectively correlate different frequency components. Unlike existing Fourier-based methods that fail to fully exploit the dependencies between low and high frequencies, FourierMamba employs a unique scanning mechanism to encode frequencies in both spatial and channel dimensions. In the spatial dimension, it uses zigzag coding to rearrange frequencies from low to high, ensuring orderly correlation. In the channel dimension, it directly applies Mamba to enhance frequency correlation and channel representation. Extensive experiments demonstrate that FourierMamba outperforms state-of-the-art methods in both qualitative and quantitative evaluations, offering a significant advancement in image deraining by better utilizing frequency information.

**Claims And Evidence:**

The author clearly express the claims made in the manuscript.

**Essential References Not Discussed:**

No

**Experimental Designs Or Analyses:**

The experimental design is well-justified, and the proposed method demonstrates competitive performance on both synthetic and real-world datasets.

**Methods And Evaluation Criteria:**

To address the insufficient utilization of correlations among different frequencies, this paper introduces Mamba combined with the Fourier transform to model the dependencies between frequencies, thereby enhancing the representation of frequency information. Experiments conducted on multiple datasets demonstrate that the proposed method achieves effective frequency correlation, showcasing its potential to break through the limitations of existing modeling frameworks.

**Other Comments Or Suggestions:**

See the weakness.

**Other Strengths And Weaknesses:**

Strengths:
1.The paper introduces a novel combination of Fourier learning and Mamba, which is well-motivated and effectively improves image deraining. It explores further possibilities of integrating Mamba with the Fourier transform.
2.The paper proposes a scanning method based on zigzag coding to systematically correlate different frequencies. This method introduces zigzag coding in the Fourier space to rearrange frequency orders, thereby orderly connecting the relationships between frequencies. The zigzag scanning strategy is well-motivated and technically sound.
3.Compared to other Mamba models for image restoration tasks, Fourier Mamba demonstrates higher efficiency and better performance, showcasing the enhancement of Fourier learning on Mamba's modeling capabilities.
4.The proposed model achieves a balance between accuracy and reasonable computational cost and model size.
5.The paper is well-organized and relatively easy to follow, with experimental results validating the effectiveness of the proposed modules and scanning methods.

Weaknesses:
1.The authors should clarify why the Fourier transform is necessary in the channel dimension instead of directly scanning the sequence.
2. There should be a proper discussion on how to ensure that normal spatial features can be obtained by inverse transformation after Fourier space scanning?
3.There are minor formatting issues, such as the need for a space before "More" in line 130.

**Questions For Authors:**

See the weakness.

**Relation To Broader Scientific Literature:**

Previous work has demonstrated the importance of frequency in image restoration tasks such as image deraining. This paper emphasizes the significance of establishing correlations between different frequencies and proposes a customized Fourier-based method for image deraining.

**Theoretical Claims:**

The effectiveness of frequency correlation is extensively validated and analyzed on a wide range of rainy datasets.

---

> ### Author Rebuttal · Authors · 2025-03-31
>
> **R1**: The Necessity of Channel-Dimensional Fourier Transform
>
> In Appendix A.6 of the paper, titled "Reasons for Using Channel-Dimensional Fourier," we note that different channels typically exhibit distinct degradation characteristics, which collectively determine the global information of an image when integrated across channels. This observation draws inspiration from style transfer research, such as the use of the Gram matrix to represent global style information [1]. Building on this insight, we introduce the Fourier transform in the channel dimension to enhance the representation of global information by capturing frequency dependencies across channels. Directly scanning the sequence (e.g., using Mamba) fails to effectively leverage the global properties inherent in the Fourier domain. In contrast, the channel-dimensional Fourier transform enables the decoupling of degradation information (e.g., rain streaks) from background content, thereby improving Mamba’s capacity to model frequency correlations. The effectiveness of this design is substantiated by ablation studies presented in Table 3 and Figure 17, which provide quantitative metrics and visualizations, respectively, demonstrating its impact.
>
> **R2**: Ensuring Normal Spatial Features After Inverse Transformation from Fourier Space Scanning
>
> You raised the concern that the paper should discuss how normal spatial features are preserved through inverse transformation following Fourier space scanning. We acknowledge that the current discussion on this aspect is insufficiently detailed and will address this in the revised manuscript by supplementing relevant content. The key to FourierMamba lies in its scanning strategy, which is designed to preserve the symmetry and global properties of the Fourier domain. Specifically, in the spatial dimension, we employ zigzag coding for scanning and process only half of the spectrum, leveraging the symmetry of the Fourier transform (amplitude centro-symmetry and phase anti-centro-symmetry) to deduce the other half (see Section 3.2). This approach ensures the integrity of the Fourier domain information. In the channel dimension, Mamba scanning is performed on a one-dimensional Fourier spectrum, adhering to similar symmetry principles. The inverse transformation relies on the standard Inverse Fast Fourier Transform (IFFT) algorithm, which theoretically guarantees perfect reconstruction from the Fourier domain to the spatial domain, provided that the scanning operation does not introduce irreversible information loss. Our experimental results (e.g., Figures 5 and 9) demonstrate that the inverse-transformed images retain normal spatial features, such as textures and details, owing to the orderly nature of the scanning design and the preservation of symmetry. We will enhance Sections 3.2 and 3.3 with these clarifications and may include a mathematical derivation in the appendix to further elucidate this process.
>
> **R3**: Formatting Issues
>
> We will address and correct spelling errors in the revised manuscript.
>
> [1] Li, Y., Fang, C., Yang, J., Wang, Z., Lu, X., and Yang, M.-H.Universal style transfer via feature transforms. Advances in neural information processing systems, 30, 2017.

---

### Decision · Program_Chairs · 2025-05-01

**Decision:**

Accept (poster)

**Comment:**

This paper proposes FourierMamba, an image deraining framework that combines Mamba-based state-space modeling with Fourier domain representations. The key contribution lies in a zigzag scanning strategy that captures inter-frequency dependencies, enhancing both spatial and channel-wise frequency modeling. The approach is well-motivated and grounded in both signal processing and recent modeling techniques.
The reviewers found the method to be original and effective, with strong empirical results across multiple datasets. The paper is clearly written and supported by thorough experiments and ablation studies. Some concerns were raised about the novelty of individual components and training consistency, but the authors addressed these points with detailed rebuttals and clarifications.

Overall, this is a well-executed and solid contribution to frequency-domain image restoration. I recommend acceptance.